# Drug-induced increase in lysobisphosphatidic acid reduces the cholesterol overload in Niemann–Pick type C cells and mice

Dimitri Moreau[1], Fabrizio Vacca[1] (iD), Stefania Vossio[1], Cameron Scott[1], Alexandria Colaco[2], Jonathan Paz Montoya[3], Charles Ferguson[4], Markus Damme[5], Marc Moniatte[3], Robert G Parton[4] (iD), Frances M Platt[2] & Jean Gruenberg[1,*] (iD)

## Abstract

Most cells acquire cholesterol by endocytosis of circulating low-density lipoproteins (LDLs). After cholesteryl ester de-esterification in endosomes, free cholesterol is redistributed to intracellular membranes via unclear mechanisms. Our previous work suggested that the unconventional phospholipid lysobisphosphatidic acid (LBPA) may play a role in modulating the cholesterol flux through endosomes. In this study, we used the Prestwick library of FDA-approved compounds in a high-content, image-based screen of the endosomal lipids, lysobisphosphatidic acid and LDL-derived cholesterol. We report that thioperamide maleate, an inverse agonist of the histamine H3 receptor HRH3, increases highly selectively the levels of lysobisphosphatidic acid, without affecting any endosomal protein or function that we tested. Our data also show that thioperamide significantly reduces the endosome cholesterol overload in fibroblasts from patients with the cholesterol storage disorder Niemann–Pick type C (NPC), as well as in liver of $Npc1^{-/-}$ mice. We conclude that LBPA controls endosomal cholesterol mobilization and export to cellular destinations, perhaps by fluidifying or buffering cholesterol in endosomal membranes, and that thioperamide has repurposing potential for the treatment of NPC.

**Keywords** endosome; lipidomics; lysosomal storage disease LSD; phospholipid; thioperamide
**Subject Categories** Membrane & Intracellular Transport; Molecular Biology of Disease

## Introduction

Most mammalian cells acquire cholesterol via the uptake of circulating LDLs. After binding to the LDL receptor or other family members, LDLs are delivered to early endosomes where they are uncoupled from their receptor [1]. While the receptor is recycled to the plasma membrane for another round of utilization, LDLs are packaged into maturing multivesicular endosomes and transported towards late endosomes and lysosomes [2]. In these late endocytic compartments, the LDL particle is disaggregated upon degradation of associated apoproteins, and cholesteryl esters are de-esterified by acidic lipase LIPA. From there, free cholesterol redistributes to intracellular membranes, including the endoplasmic reticulum where it controls the expression of cholesterol-dependent gene via the SREBP pathway [3,4]. However, the mechanisms that regulate the endosomal cholesterol content, control its translocation across the endosomal membrane and further export towards other destinations are not clear. When mutated, two genes NPC1 and NPC2 are responsible for the cholesterol storage disease Niemann–Pick type C, but the functions of the NPC1 and NPC2 proteins remain incompletely understood [5].

We previously observed that in NPC cells, cholesterol accumulates in late endosomes containing lysobisphosphatidic acid (LBPA) [6,7], also referred to as bis(monoacylglycero)phosphate (BMP). This atypical phospholipid accounts for approximately 15 Mol% of total phospholipids in the multivesicular late endosomes of BHK cells, where it abounds within intraluminal membranes, and is not detected in other subcellular compartments [7,8]. In addition to this uncommon distribution, LBPA also exhibits an unusual stereo-configuration [9,10], and its metabolism is poorly understood [11,12]. After de-esterification, free LDL-derived cholesterol is incorporated into membrane of LBPA-containing late endocytic compartments. In fact, our data indicate that LBPA plays a direct role in regulating the cholesterol flux through endosomes. Interfering with

1 Department of Biochemistry, University of Geneva, Geneva 4, Switzerland
2 Department of Pharmacology, University of Oxford, Oxford, UK
3 Mass Spectrometry Core Facility, EPFL, Lausanne, Switzerland
4 Institute for Molecular Bioscience and Center for Microscopy and Microanalysis, University of Queensland, Brisbane, Qld, Australia
5 Biochemisches Institut, Christian-Albrechts-Universität, Kiel, Germany
*Corresponding author. Tel: +41 2237 9 3464; E-mail: jean.gruenberg@unige.ch

LBPA functions causes cholesterol accumulation in late endosomes, phenocopying NPC [6,13], and inhibits Wnt-induced biogenesis of lipid droplets [14]. In this paper, we describe a strategy to identify compounds that influence levels and/or distribution of LBPA. We identify thioperamide as a novel modulator of LBPA levels in late endosomes and show that it decreases the cholesterol overload in fibroblasts from NPC patients and in NPC null mice.

# Results

### High-content screen to identify compounds that modulate LBPA

Our initial goal was to identify compounds that influence LBPA using a monoclonal antibody against LBPA in a high-content image-based screen [7]. Since free LDL-derived cholesterol is released in late endosomes containing LBPA (Fig 1A), and given our observations that cholesterol is functionally linked to LBPA [6], we also monitored cholesterol, using the polyene macrolide filipin, which binds cholesterol and conveniently emits in the UV range. As additional markers for cell segmentation during automated image analysis, we used propidium iodide to label nuclei and cell tracker to label the cytoplasm.

The screen was carried out with the Prestwick library of FDA-approved drugs (http://www.prestwickchemical.com/libraries-screening-lib-pcl.html; and see Datasets EV1 and EV2), because of high chemical and pharmacological diversity, low toxicity, known bioavailability and structures, and also because some target data are available. Figure 1B shows an example of a 384-well plate stained for cholesterol (green) and LBPA (red) with four micrographs stitched together per well. As negative controls, all wells in one column were treated with DMSO only (Fig 1B, left of the plate), and as positive controls, all wells in another column were treated with the pharmacological agent U18666A (Fig 1B, right of the plate), a charged sterol analog that mimics NPC [15], binds the NPC1 protein [16] and causes the accumulation of both cholesterol and LBPA in late endosomes [6,14] (Fig 1C). The broad dynamic range of the detection system and its robustness, as already revealed in our previous RNAi screens with the same detection system [14], are illustrated by the staining with both markers in U18666A-treated versus control cells (Fig 1C and D).

While most compounds had no effect on the two lipids as anticipated, the majority of the compounds that did have an effect caused

a concomitant increase in both cholesterol and LBPA, much like U18666A (Fig 1D). However, a few compounds caused an apparent increase in LBPA staining with minimal effect on cholesterol levels—a phenotype particularly striking in cells treated with thioperamide maleate (Fig 1B lower well in the boxed area, and Fig 1D, pink dot). We thus decided to focus on thioperamide and to further characterize the effects of this compound.

### Effects of thioperamide on LBPA levels

The selective increase in LBPA, but not cholesterol, is well illustrated in high-magnification views of the cells treated with thioperamide (Fig 1D and E). Automated, unbiased quantification of the screen data confirmed that LBPA staining intensity was increased highly significantly in thioperamide-treated cells without any visible toxic effect, while both LBPA and cholesterol levels increased in the presence of U18666A (Fig 1C–E) as expected. Similarly, trimeprazine caused a marked increase in LBPA (Fig 1C–E; light blue in Fig 1D), but also raised cholesterol levels to the same extent as U18666A (Fig 1D and E).

In addition, thioperamide affected equally different cells types of human or rodent origin (Fig 2A). Consistent with our imaging data (Fig 1), a biochemical analysis confirmed that, in marked contrast to U18666A, thioperamide did not affect the levels of free cholesterol, esterified cholesterol and total cholesterol, when compared to controls (Fig 2B). Moreover, automated unbiased quantification further demonstrated that the cholesterol content of individual LBPA endosomes after thioperamide treatment was very similar to the DMSO controls—and also to most cells treated with Prestwick compounds (Fig 2C). Finally, an analysis by electron microscopy showed that the ultrastructure of individual endosomes looked very similar in thioperamide-treated cells when compared to controls (Fig EV1A). However, immunogold labelling of cryosections using the anti-LBPA antibody confirmed that the amounts of LBPA in multivesicular late endosomes increased in thioperamide-treated cells (Fig 3A, pseudo-coloured individual endosomes and uncoloured originals in Fig EV1B; double-blind quantification in Fig 3B).

While differences in staining intensity alone already revealed major differences in the effects of some compounds, the automated image analysis pipeline captured additional parameters (e.g. size, area, integrated intensity, average intensity and object count), to describe and compare the various staining patterns.

**Figure 1. High-content screen using the Prestwick library to identify compounds that modulate endosomal lipids.**

A   Outline of endocytic cholesterol transport. After endocytosis by the LDL receptor, LDL is delivered to early and then late endosomes. Free cholesterol is released in late endosomes and then exported to its cellular destinations.

B   Example of a 384-well plate treated with compounds. The panel shows the staining with filipin (free cholesterol in green pseudo-colour) and anti-LBPA antibodies (red). The leftmost column shows control wells treated with DMSO only, and the right penultimate column shows wells treated with U18666A.

C   Effects of U18666A, thioperamide and trimeprazine, compared with controls. Cells treated with the indicated compounds or with DMSO alone for 18 h at 37°C were processed for immunofluorescence microscopy. Low-magnification views are shown after staining with anti-LBPA antibodies (red), filipin (green pseudo-colour) and DAPI (blue). The scale bar is 20 μm.

D   Screen data plot. The plot shows the integrated intensity of LBPA versus filipin (cholesterol) staining in all HeLa cells of each well analysed in the screen (each dot is the average of replicates in the duplicate plates, and ≥ 600 cells were analysed per compound). Only compounds that showed less than 20% toxicity are plotted. The cells treated with U18666A, thioperamide maleate and trimeprazine tartrate are indicated, as well as the DMSO controls.

E   Integrated intensity of cholesterol and LBPA staining. The integrated intensity of cholesterol (filipin in green) and LBPA (red) staining in all cells of the indicated samples is quantified after normalization to the DMSO controls ($n$ = 3 independent experiments, 64 images analysed per experiments, error bars = SD, two-way ANOVA; **$P$ < 0.01; ***$P$ < 0.005).

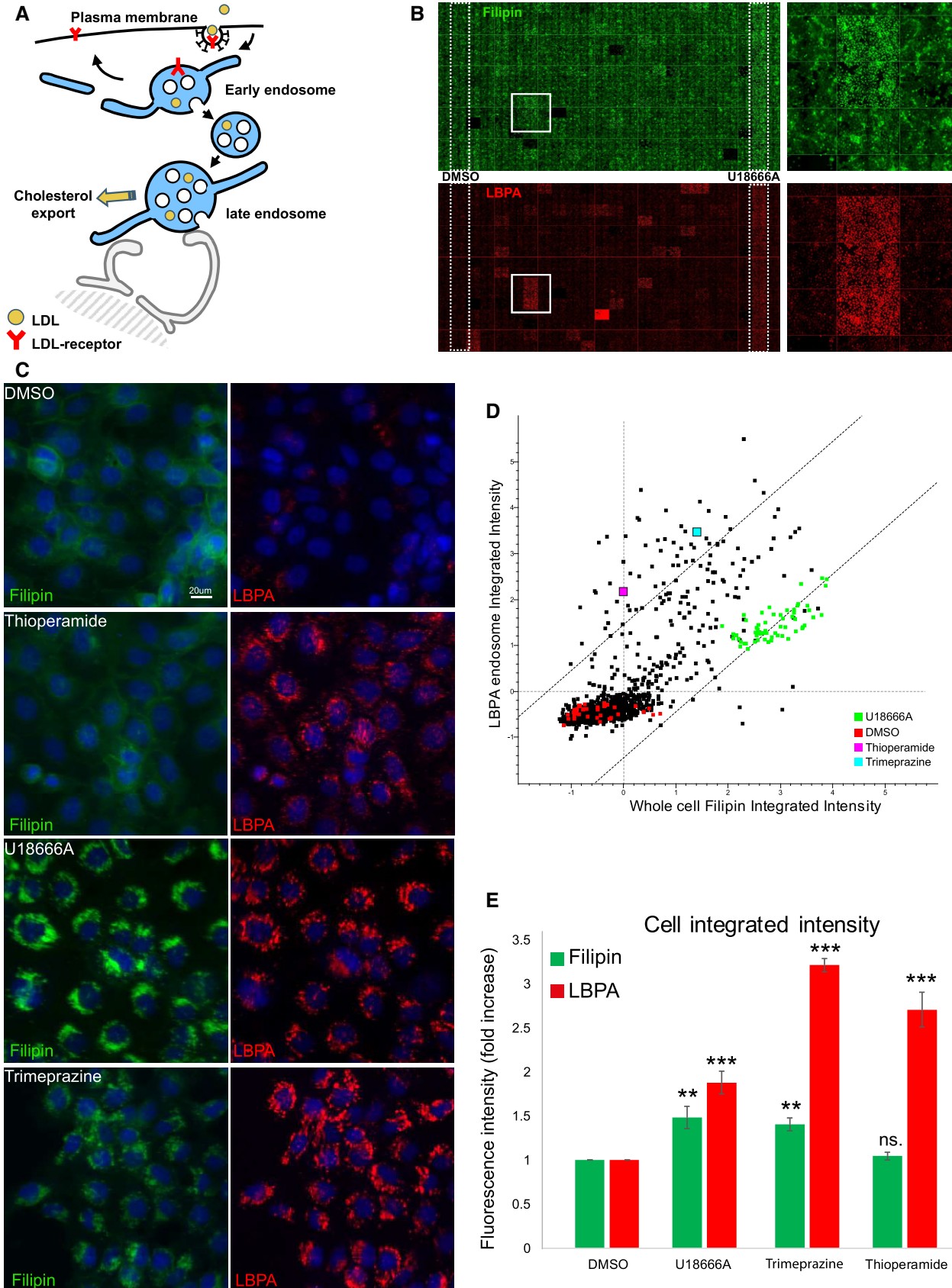

**Figure 1.**

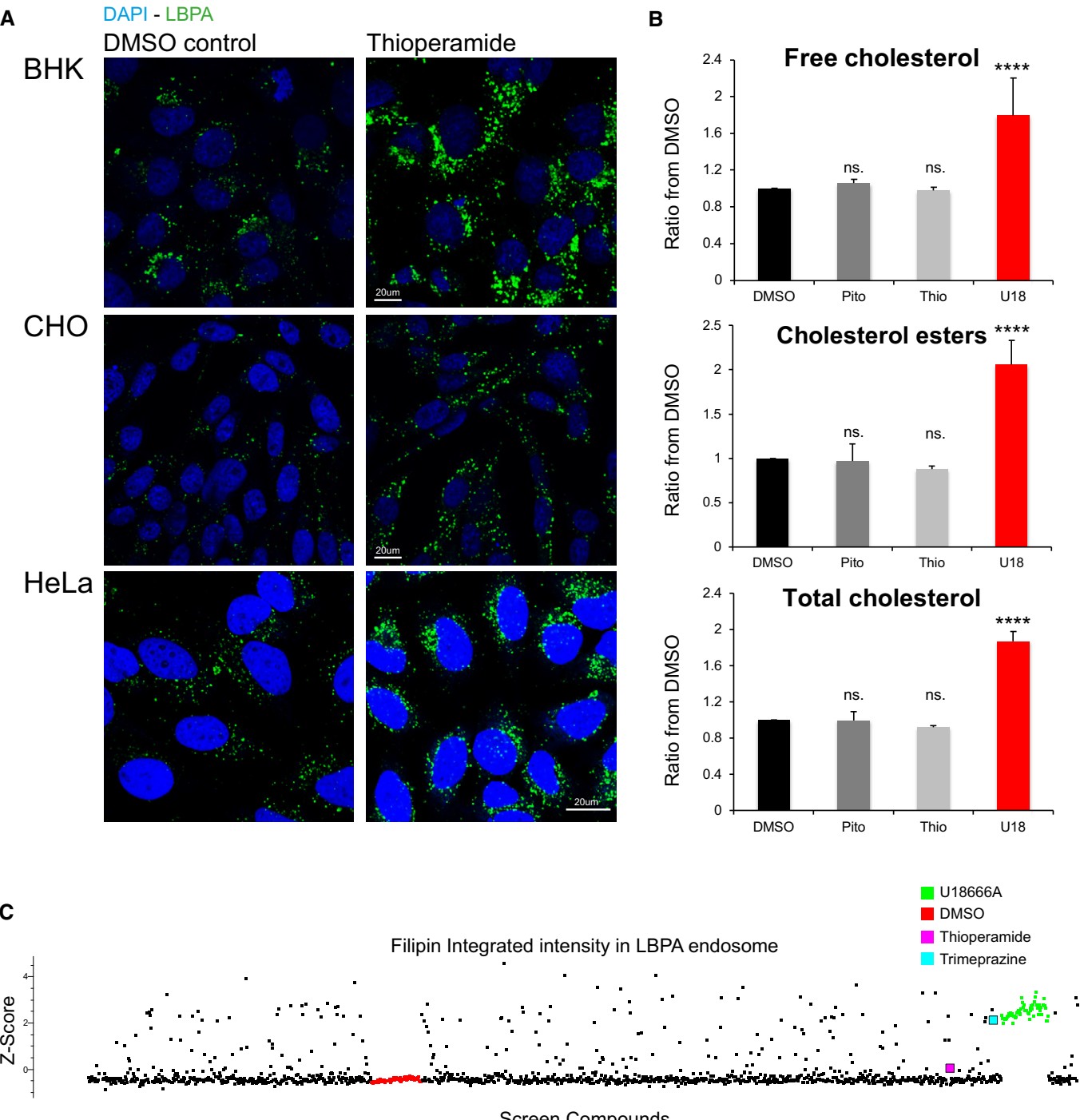

**Figure 2. Thioperamide affects LBPA but not cholesterol.**

A Thioperamide treatment of tissue culture cells. BHK, CHO and HeLa cells were treated with thioperamide and processed as in Fig 1C. Low-magnification views are shown after staining with anti-LBPA antibodies (green) and DAPI (blue). *n* = 3 independent experiments with > 200 images acquired and analysed automatically.

B Cholesterol quantification by mass spectrometry. A431 cells were treated with DMSO alone, pitolisant 10 µM (Pito), thioperamide 10 µM (thio) or U18666A 10 µM (U18) for 18 h at 37°C. After extraction, free cholesterol, cholesteryl esters and total cholesterol were quantified by mass spectrometry and normalized to the DMSO controls (*n* = 3 independent experiments, error bars = SD, one-way ANOVA, ****$P$ < 0.0001).

C Screen data plot of the cholesterol content of LBPA endosomes. Automated unbiased quantification of the filipin integrated fluorescence signal (cholesterol content) in LBPA-containing endosomes, after treatment with each compound of the Prestwick library. As in Fig 1D, each dot is the average of replicates in the duplicate plates, and ≥ 600 cells were analysed per compound—only compounds that showed < 20% toxicity are plotted. The fluorescence signal of LBPA endosomes was used to segment the imaged and generate a mask, which was then applied on the micrographs to quantify the integrated intensity of filipin staining. The samples treated with thioperamide (pink), trimeprazine (light blue), DMSO (red) and U18666A (green) are indicated.

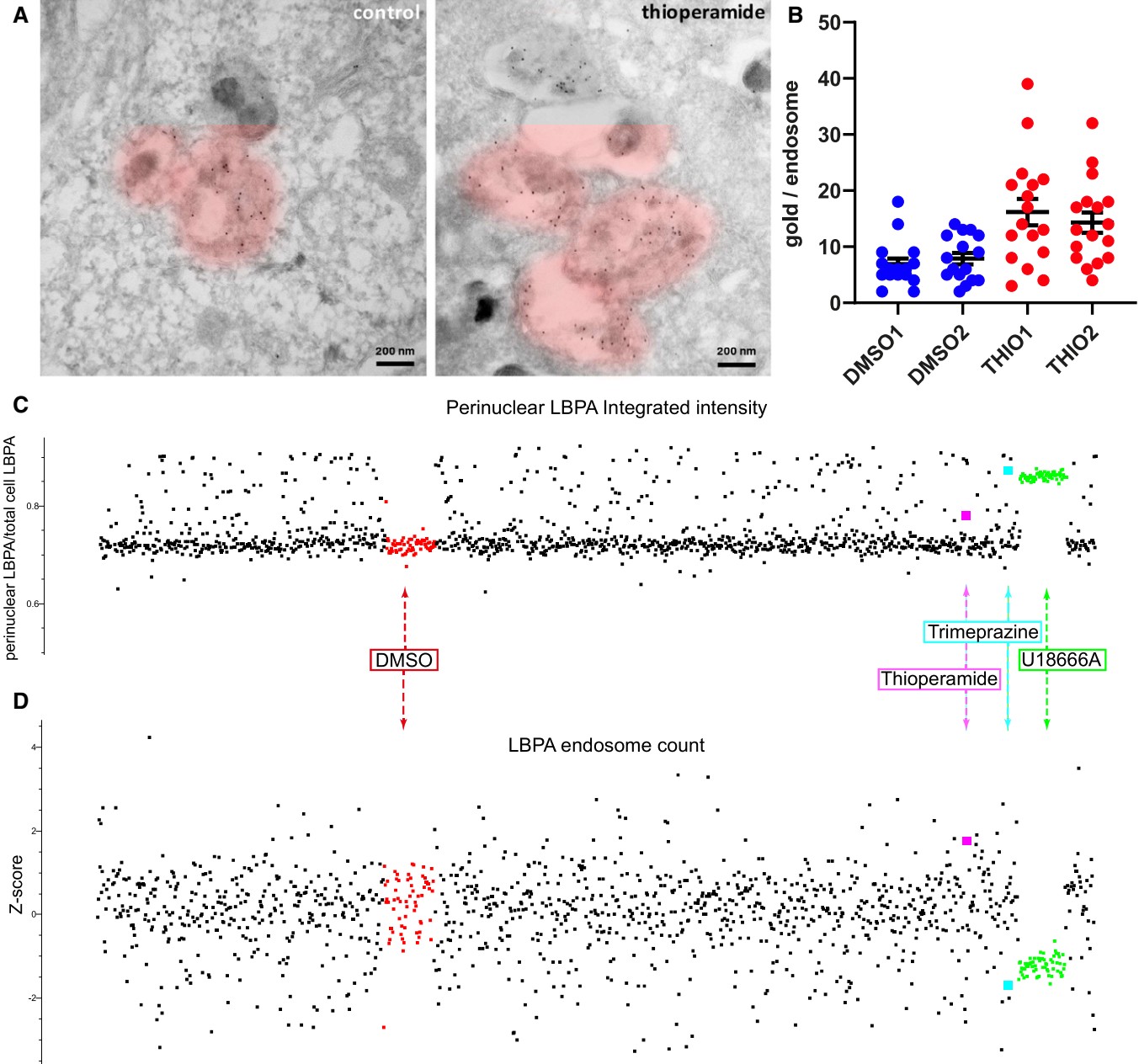

**Figure 3. Thioperamide does not affect endosome morphology or distribution.**

A, B   Immunogold labelling with anti-LBPA antibodies. HeLa MZ cells were treated or not with thioperamide for 18 h and processed for electron microscopy. Cryosections were labelled with antibodies against the late endosomal lipid LBPA followed by 5 nm protein A-gold. Multivesicular endosomes are pseudo-coloured (see Fig EV1 for uncoloured images). Scale bars, 200 nm. The data in (A) were quantified in a double-blind analysis of two sets of 16 micrographs for each condition; the number of gold particles per endosome is shown in a scatter plot (B) for each endosome identified without any bias in each micrograph of the two control (DMSO) and thioperamide-treated (THIO) samples.

C   Distribution of LBPA-containing endosomes in the perinuclear region. Automated unbiased quantification of LBPA fluorescence in the perinuclear region of the cells, after treatment with each compound of the Prestwick library as in Fig 2C. As in Fig 1D, each dot is the average of replicates in the duplicate plates, and ≥ 600 cells were analysed per compound—only compounds that showed < 20% toxicity are plotted. The integrated intensity of the LBPA fluorescence signal was measured within a mask of the perinuclear region in each cell, calculated from the nuclei DAPI staining. The samples treated with thioperamide (pink), trimeprazine (light blue), DMSO (red) and U18666A (green) are indicated.

D   Total number of LBPA endosomes. The number of individual LBPA-positive structures was measured in the whole cell. The z-factors are shown to evaluate the distribution of endosomes containing LBPA; colour code as in (C).

Given the large set of parameters, the combination of variables was transformed into principal components to best explore the phenotypic space. The principal component analysis (PCA) confirmed that most compounds had no significant effect on cholesterol and LBPA, while all U18666A-treated cells (green, Movie EV1) clustered together far away from the bulk (white, Movie EV1). Interestingly, the PCA also demonstrates that the thioperamide phenotype (pink, Movie EV1) is set far away from all other compounds, including the positive control U18666A (green, Movie EV1) or any compound affecting LBPA or cholesterol. These observations further confirm the notion that the cellular effects of thioperamide are unique and remarkable.

### Effects of thioperamide on endosome distribution

The differences observed in the PCA space between thioperamide versus U18666A or trimeprazine were due to differences not only in the cholesterol content of LBPA-containing endosomes, but also in their subcellular distribution. After U18666A or trimeprazine treatment, LBPA-positive late endosomes loaded with cholesterol were typically clustered in the perinuclear region (Fig 1C). This distribution agrees well with previous observations by us [13] and others [17,18] that cholesterol accumulation in NPC cells paralyses late endosomes at the microtubule minus ends, close to the nucleus. In marked contrast, thioperamide did not seem to cause perinuclear clustering of LBPA endosomes (Fig 1C), in line with our observations that it did not affect cholesterol (Figs 1B–E and 2B). Automated unbiased quantification of all cells treated with Prestwick compounds confirmed that U18666A and trimeprazine caused a highly significant increase in perinuclear LBPA staining (Fig 3C). By contrast, thioperamide had essentially no effect on endosome distribution, much like DMSO in controls or most Prestwick compounds (Fig 3C). In addition, counting the number of individual LBPA-positive structures after treatment with Prestwick compounds showed that thioperamide was present on the side of the distribution opposite to trimeprazine or U18666A (z-factors shown in Fig 3D)—the distribution is relatively noisy because of cell-to-cell variation. Indeed, LBPA endosomes clustered in the vicinity of the nucleus after trimeprazine or U18666A treatment were no longer well resolved, resulting in an apparent decrease in the number of labelled endosomes.

### Effects of thioperamide on endosome functions

We then investigated whether thioperamide had any effect on other endosomal functions. No change was observed in the amounts of markers of early (EEA1, transferrin receptor) or late (LAMP1, CD63) endocytic compartments (Fig EV2A). Neither was the distribution of these markers changed by the drug (Fig EV2A) —the clustering effects of U18666A are shown for comparison. Similarly, thioperamide did not affect the endolysosome acidification capacity or the number of acidic endolysosomes (Fig EV2C) —the effects of the V-ATPase inhibitor bafilomycin A1 are shown after a short (2 h) treatment for comparison. During infection with vesicular stomatitis virus (VSV), the release of viral RNA into the cytoplasm depends on functionally intact endosomes [19,20] and is inhibited by cholesterol accumulation in endosomes [21]. After incubating cells for 3 h with recombinant VSV expressing

GFP-tagged P-protein [22,23] at a low physiologically relevant MOI (1.0), no difference could be observed between thioperamide- and mock-treated control cells (Fig EV2B). Moreover, the degradation of the epidermal growth factor (EGF) receptor in cells challenged with EGF occurred with identical kinetics in cells treated with thioperamide and in controls (Fig EV2D; quantification by automated microscopy shown in the bar graph). Altogether, these data demonstrate that thioperamide increases LBPA levels in late endosomes highly selectively without affecting their cholesterol content and that endosomes and lysosomes of thioperamide-treated cells are functionally intact.

### Thioperamide targets the histamine receptor HRH3

Thioperamide is reported to act as an inverse agonist of the histamine receptor H3 (HRH3) [24] but it also acts on HRH4, another member of the same receptor family [25]. HRH3 is closely related to HRH4 (37% sequence identity) and more distantly related to HRH1 or HRH2 (21% sequence identity), the two other members of this family. We thus compared the effects of compounds known to target histamine receptors, including those that were already present in the Prestwick library, as well as a small library of additional HRH3/HRH4 antagonists and histamine analogs (Dataset EV3). Strikingly, 10 out of 12 compounds targeting HRH3 or HRH4, which all act as antagonists or inverse agonists, exhibited a thioperamide-like increase in LBPA without changing cholesterol levels (Fig 4A, Dataset EV3). These observations strongly argue against the notion that the increase in LBPA levels was caused by some off-target effects of thioperamide, since the chemical scaffolds of these compounds are different. In marked contrast, LBPA accumulation was observed with only 2 out of the 26 compounds against HRH1 and with none of the five compounds that target HRH2 (Fig 4A). Neither did histamine (not shown) nor any histamine receptor agonist was tested (Fig 4A and Dataset EV3).

We also found that pitolisant, which was not in the Prestwick library (Dataset EV3), increased LBPA levels to a similar extent as thioperamide, without affecting cholesterol levels (Figs 2B and 4B), again much like thioperamide. Moreover, the dose–response profiles (Fig EV3A) and time courses (Fig EV3B) of LBPA accumulation were similar with both thioperamide and pitolisant. However, treatment with pitolisant reduced the cell number at long time-points (Fig EV3C), suggesting that the drug may have some cytotoxic effects. We thus centred our efforts on thioperamide alone.

Further support for the notion that thioperamide targets HRH3 as an inverse agonist came from the observations that levels of LBPA and HRH3 were inversely correlated. Indeed, in a mixed population of cells expressing GFP-tagged HRH3 (Fig 4C), the cellular intensity of the LBPA signal is skewed towards cells expressing low levels of HRH3-GFP and vice versa (Fig 4D), indicating that LBPA levels are low in cells expressing high HRH3-GFP levels and high in cells expressing low HRH3-GFP levels. These observations were confirmed using cells stably expressing HRH3-GFP. In these cells, HRH3-GFP could be efficiently depleted after knockdown with four different siRNAs (Fig 4E and F, quantification in Fig 4G). Again, LBPA and HRH3-GFP levels were anti-correlated, and HRH3-GFP depletion was accompanied with a concomitant increase in LBPA levels (Fig 4F and G).

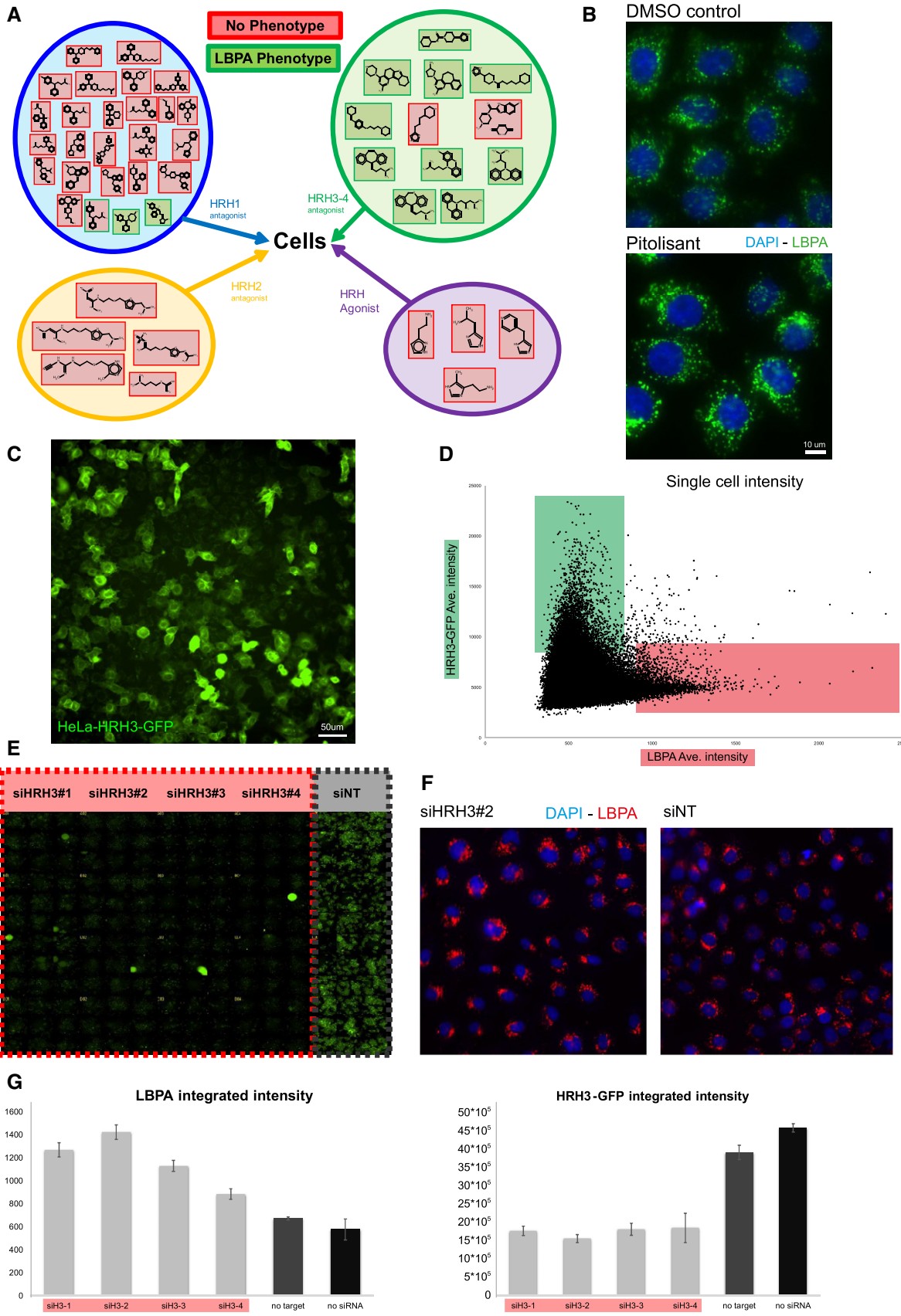

**Figure 4.**

◀

**Figure 4. Thioperamide targets the histamine receptor HRH3.**

A   Effects of compounds that target histamine receptor family members. The outline shows the compounds (listed in Dataset EV3) that target histamine receptor family members as antagonists (HRH1, HRH2, HRH3 and HRH4) or agonists. These include the compounds originally present in the Prestwick library as well as additional compounds selected from the literature. Cells treated with this small library of compounds were analysed by automated microscopy after labelling with DAPI and antibodies against LBPA. Compounds that increased LBPA staining intensity, such as thioperamide (Fig 1) or pitolisant (Fig 4B), are green, while compounds without effects are red (n = 2 independent experiments with 96 images acquired and analysed automatically).

B   Effect of pitolisant on LBPA. HeLa MZ cells treated or not with pitolisant 10 μM for 18 h at 37°C were labelled with DAPI and anti-LBPA antibodies and analysed by fluorescence microscopy (quantification in Fig EV3A).

C, D  LBPA intensity in a mixed population of cells expressing HRH3-GFP. After transfection with HRH3-GFP, uncloned stably expressing cells (C) were labelled with anti-LBPA antibodies and analysed by automated microscopy (C). Unbiased quantification (D) shows the inverse correlation between HRH3-GFP expression (high expressing cells in green) and the endosome integrated intensity of LBPA staining (high LBPA labelling in red) (n = 2 independent experiments with 500,000 cells analysed automatically, error bars = SD).

E–G  LBPA intensity in cells expressing or not HRH3-GFP. HeLa MZ cells stably expressing HRH3-GFP grown in 96-well plates were separately treated, with four different siRNAs that target HRH3 (siHRH3#1; siHRH3#2; siHRH3#3; and siHRH3#4) or with control non-target (siNT) siRNA (E). HRH3-GFP was analysed by fluorescence microscopy in four columns of cells per condition; micrographs are stitched together in the montage. Panel (F) shows an example of cells treated with siHRH3#2 or siNT, and then labelled with DAPI and antibodies against LBPA. Cells treated as in (E) and labelled with DAPI and antibodies against LBPA as in (F) were analysed by automated microscopy, and the integrated intensities of LBPA (left panel) and HRH3-GFP (right panel) signals are compared (G) (n = 2 independent experiments with 192 images acquired and analysed automatically, error bars = SD).

## Thioperamide reduces cholesterol overload in NPC cells

Cholesterol accumulates in late endocytic compartments of NPC cells, eventually leading to a pathological enlargement of these compartments—a characteristic of lysosomal storage disorders. Concomitant with this increase in endosomal volume, protein and lipid [21], total LBPA levels are also increased [6]. The accumulation of storage materials in NPC endosomes eventually leads to a traffic jam and a collapse of endosomal membrane dynamics [26,27]. Given the role of LBPA in endosomal cholesterol transport [6,14,28], we reasoned that the capacity of LBPA to accommodate—or buffer—excess cholesterol may eventually become limiting in NPC endosomes.

To test the effects of thioperamide in NPC cells, we used three fibroblast lines obtained from patients with well-established mutations in the NPC1 or NPC2 gene. In all three cell lines, thioperamide caused after 72 h a very significant decrease in cholesterol levels (Fig 5A, quantification in Fig 5B at 72 h). In contrast to control cell lines (Figs 1 and 2A), LBPA levels were also decreased in NPC cell lines after a 72-h treatment (Fig 5B). This decrease was not due to some toxic effects of the compound in NPC1 or NPC2 cells after 48 h (Fig 5D) or 72 h (Fig 5E). If anything, the number of thioperamide-treated NPC cells after 72 h was somewhat higher than the controls, suggesting that drug may increase cell survival. We therefore hypothesized that the effects of the drug on LBPA levels may be obscured in NPC cells by the beneficial decrease in storage of cholesterol and other lipids including LBPA. Indeed, a significant increase in LBPA levels was transiently observed at a shorter timepoint, before changes in cholesterol could be detected (Fig 5B at 48 h)—this increase is particularly striking, since even before thioperamide addition, LBPA levels are already much higher in NPC cells when compared to controls [6] (see also LBPA quantification in NPC1 mice, Fig 6B). The simplest interpretation may be that, much like in other cell types, thioperamide increases LBPA levels in NPC cells, thus facilitating cholesterol mobilization, consistent with previous findings [28], which in turn reduces storage materials in endosomes and restores physiological levels of LBPA and other lipids.

An analysis by mass spectrometry showed that total cellular cholesterol normalized to total cellular lipids was also reduced by thioperamide treatment of all three NPC cell lines (Fig 5C). At the total cellular level (Fig 5C), the effects of the drug appear less prominent than at the endosomal level (Fig 5B), because in NPC fibroblasts, the amounts of cholesterol accumulated in endosomes correspond only to a fraction of total cellular cholesterol [29]. We compared the reduction in cholesterol levels to treatment with cyclodextrin, currently considered to be one of the—if not the—most efficient protocol to reduce cholesterol levels in cultured cells [5,30]. Cyclodextrin also decreases the progression of neurological disorders in NPC1 patients after intrathecal injection [31]. Remarkably, thioperamide reduced cholesterol levels as efficiently as cyclodextrin (Fig 5C). In addition, the feedback transcriptional regulation of cholesterol metabolism via the sterol regulatory element-binding protein (SREBP) pathway is defective **i**n NPC cells [32–34] (see also Fig 6A). To test the effects of thioperamide, we used NPC1 and NPC2 knockout cells generated using CRISPR/Cas9 [35]. Much like cyclodextrin [35], thioperamide was able to partially correct the defect in the transcriptional regulation of two canonical cholesterol-dependent genes, the LDL receptor and HMG CoA reductase in these NPC1 and NPC2 KO cells (Fig 6A). By contrast, thioperamide treatment did not affect the expression levels of proteins involved in the transfer of cholesterol from endosome to the ER or in endosome-ER membrane contact sites (Fig EV4), including FYCO1 [36], ANXA1 [37], STARD3/MLN64 [38], ORP1l, VAPA and VAPB [39,40].

## Effects of thioperamide in $Npc1^{-/-}$ mice

Finally, we tested the effects of thioperamide in mice lacking NPC1. As anticipated, the liver of $Npc1^{-/-}$ mice showed significant accumulation of cholesterol, approximately 20× higher than livers from WT littermates (Fig 6B). LBPA was quantified by mass spectrometry after efficient separation from phosphatidylglycerol (PG), because the two lipids are isobaric (Appendix Fig S1). Much like cholesterol, LBPA also accumulated approximately 10X (Fig 6B). Interestingly, the mass spectrometry analysis also revealed the presence of a triple acylated version of LBPA, semilysobisphosphatidic acid (sLBPA) [41], which could be well separated from LBPA and PG (Appendix Fig S1). sLBPA also accumulated significantly, increasing from < 0.05% of total phospholipids in WT liver to 0.8% in $Npc1^{-/-}$ liver (Appendix Fig S2A)—a value close to the physiological levels

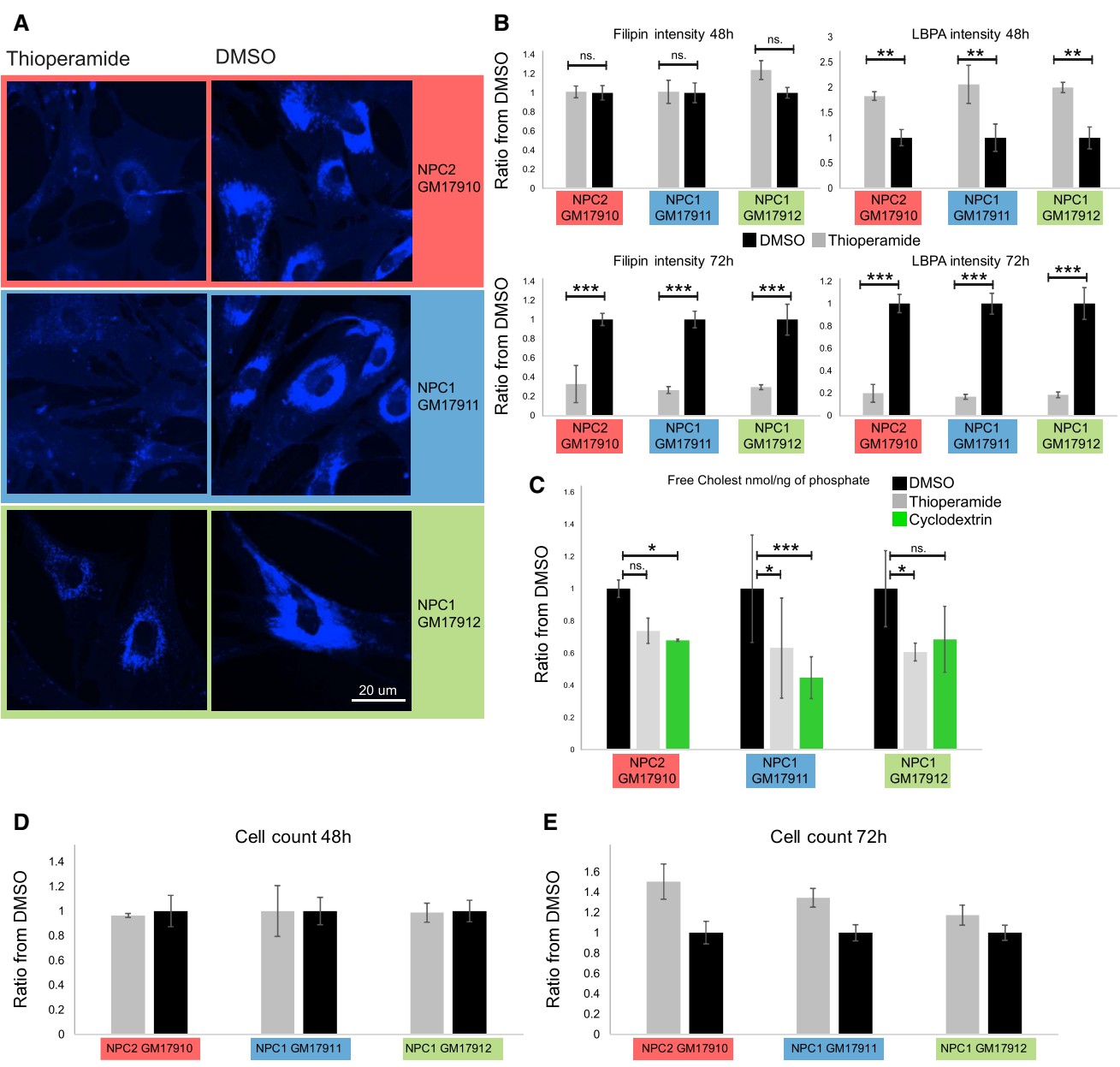

**Figure 5. Effects of thioperamide in NPC cells.**

A   Treatment of NPC fibroblasts and NPC null mice with thioperamide. Fibroblast lines obtained from patients with well-established heterozygote mutations in the NPC1 (GM17912 line: NPC1 P1007A/T1036M; GM17911 line: NPC1 I1061T/T1036M) or with a homozygote mutation in the NPC2 gene (GM17910 line: C93F/C93F) were treated or not for 72 h with thioperamide 10 μM, stained with filipin (cholesterol) and analysed by fluorescence automated microscopy.

B   Quantification of cholesterol and LBPA staining in NPC fibroblasts. Cells as in (A) were treated or not with thioperamide for 48 h or 72 h, labelled with filipin (cholesterol) and anti-LBPA antibodies, and analysed by fluorescence automated microscopy. The panels show the integrated intensity of the filipin (left) and LBPA (right) signals were quantified after 48 (top) and 72 h (bottom). The colour code of each fibroblast cell line is as in (A) (n = 3 independent experiments with 144 images acquired and analysed automatically, > 2,000 cells per experiment, error bars = SD, one-way ANOVA, *P < 0.05; **P < 0.005;***P < 0.001).

C   Quantification of cholesterol in NPC fibroblasts by mass spectrometry. NPC cell lines (A) were treated or not with thioperamide or 0.1% (2-hydroxypropyl)-β-cyclodextrin for 72 h. After extraction, lipids were analysed and quantified by mass spectrometry (n = 3; error bars = SD, one-way ANOVA, *P < 0.05; **P < 0.005; ***P < 0.001).

D, E   Quantification of cell counts in the three NPC cell lines treated with thioperamide. The experiment was as in (A-C) except that nuclei were labelled with DAPI and quantified by automated microscopy. Data are normalized to DMSO control (n = 2 independent experiments, 75 images per experiments).

of LBPA in WT liver (1.0% of total phospholipids). Moreover, our lipidomic analysis revealed a dramatic remodelling of the acyl chain composition of both LBPA (Fig EV5A) and sLBPA (Fig EV5B) in

$Npc1^{-/-}$ mouse liver, characterized by shorter acyl chains and a reduced number of double bonds, confirming the notion that a metabolic relationship exists between LBPA and sLBPA [41]. In

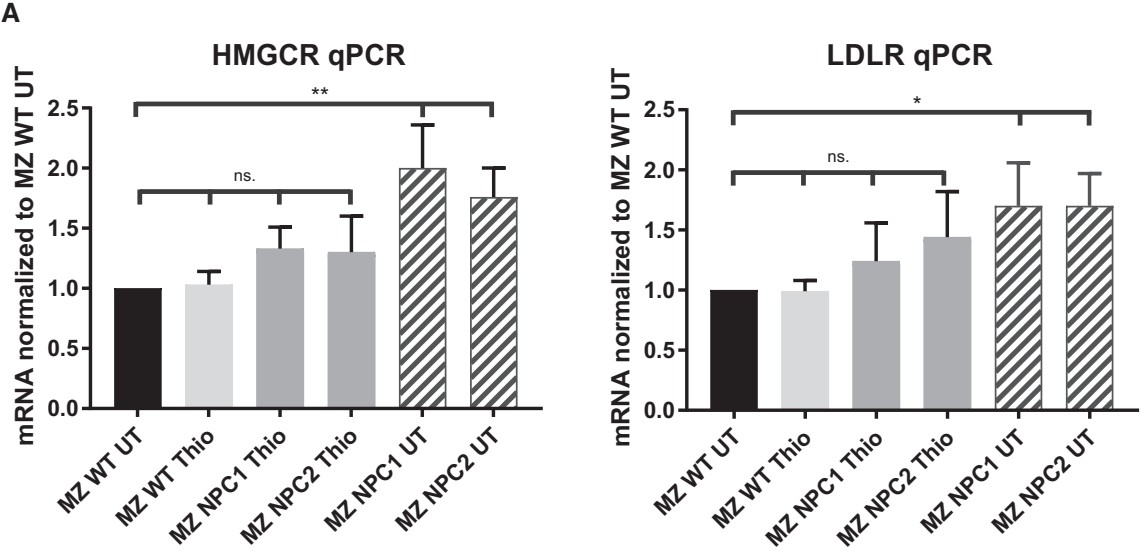

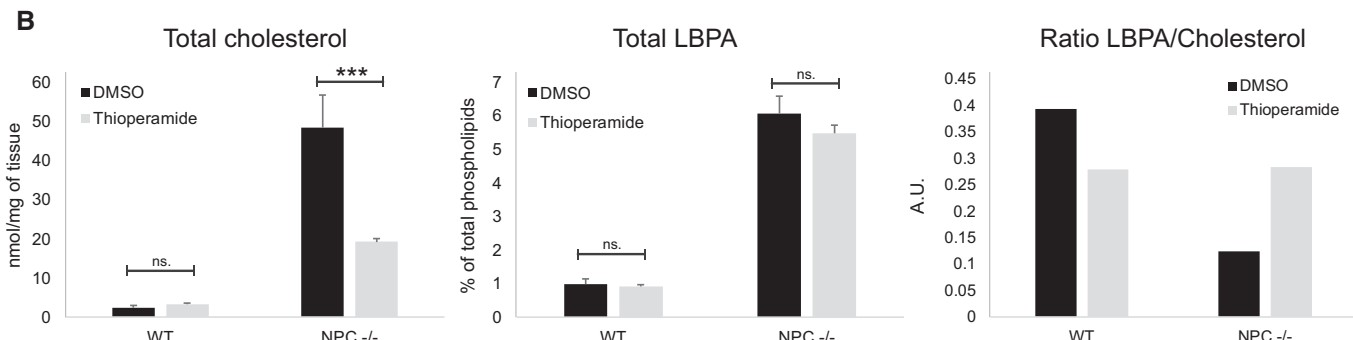

**Figure 6. Effects of thioperamide in NPC cells and mice.**

A   Effects of thioperamide on cholesterol-dependent transcriptional regulation. The parental HeLa MZ cells, NPC1 KO cells or NPC2 KO cells were treated or not with thioperamide 10 μM for 18 h. Total mRNA was extracted, and both LDLR mRNA and HMG CoA reductase mRNA were quantified by RT–PCR (n = 3 independent experiments, error bars = SD, one-way ANOVA, *P < 0.05; **P < 0.005).

B   Quantification of cholesterol in NPC null mice by enzymatic analysis. WT or NPC1 null mice were treated or not with thioperamide for 6 weeks after weaning and sacrificed. The total content of unesterified cholesterol in the corresponding liver extracts was then quantified using an enzymatic assay and is expressed in nmol per mg tissue. The total content of LBPA in the same liver extracts was quantified by mass spectrometry (see Appendix Fig S1) and is therefore expressed as a percentage of total phospholipids. The total LBPA content of WT mice was only marginally increased by the drug, either because the doses were low, or because the contribution from tissue material unaffected by the drug obscured selective changes at the cellular level (despite much effort, we were unable to visualize LBPA in liver samples by immunocytochemistry). The relative ratio of LBPA versus cholesterol values is shown.

marked contrast to LBPA and sLBPA, the total relative amounts of other phospholipids were not significantly affected in *Npc1*$^{-/-}$ mouse liver (Appendix Fig S2B). Neither was the acyl chain composition of other phospholipids, as illustrated with phosphatidylcholine (Appendix Fig S3A) and phosphatidylethanolamine (Appendix Fig S3B). These observations further strengthen the notion that LBPA and cholesterol are closely linked functionally.

WT and *Npc1*$^{-/-}$ mice, which model the most aggressive early-onset form of the disease [42], were then treated with thioperamide. The treatment started at weaning (3 weeks of age) and continued until they reached the humane end-point (loss of 10% body weight within 24 h), to minimize suffering. At that point, thioperamide did not significantly improve the life span, motor function/rearing or

high-frequency tremor of *Npc1*$^{-/-}$ mice, although some benefits were observed when combined with Miglustat (Appendix Fig S4A–D)—the only compound available as a treatment against NPC in Europe, but not in the United States, which prolongs life but does not arrest disease progression [43]. LBPA was marginally decreased in drug-treated *Npc1*$^{-/-}$ liver (Fig 6B), and sLBPA unaffected (Appendix Fig S2A), while the acyl chain composition remained unchanged (Appendix Fig S3A and B). Strikingly, however, the cholesterol levels in the *Npc1*$^{-/-}$ mouse liver were significantly reduced by the thioperamide treatment (Fig 6B), as in NPC fibroblasts (Fig 5C). In contrast to liver, total cholesterol levels in the brain of *Npc1*$^{-/-}$ mice were similar to WT (Appendix Fig S5A), as expected [44,45], perhaps reflecting the asymmetric distribution of

cholesterol in NPC neurons [46]. However, much like in liver, LBPA amounts (Appendix Fig S5B) and acyl chain composition (Appendix Fig S5C) were significantly changed. Altogether, these data suggest that in NPC cell lines and $Npc1^{-/-}$ liver cells thioperamide facilitates cholesterol mobilization, presumably by transiently increasing LBPA levels, similar to other cells (Figs 1–3), which in turn reduces storage materials in endosomes, including LBPA and other lipids. The distribution of LBPA versus cholesterol in $Npc1^{-/-}$ mice may well reflect the end-point balance between thioperamide-induced increase in LBPA and the beneficial effects of the treatment on the compartment overload. In any case, our data demonstrate that thioperamide reduces the cholesterol storage phenotype both in NPC fibroblasts and in $Npc1^{-/-}$ mice.

## Discussion

In conclusion, we find that LBPA levels (i) are increased by thioperamide and other inhibitors or inverse agonists of the HRH3 receptor that do not share a common backbone and (ii) are inversely correlated with the levels of GFP-HRH3 expression. We also find that thioperamide does not affect any endosomal protein or function that was tested, including acidification, lysosomal degradation and enveloped virus infection. Finally, we find that thioperamide decreases cholesterol overload in both NPC1 and NPC2 mutant cells as well as in $Npc1^{-/-}$ mouse liver. We did not observe significant behavioural improvement in thioperamide-treated $Npc1^{-/-}$ mice, except in combination with Miglustat. It will be interesting in the future to test the effects of thioperamide alone or in combination with Miglustat in mice expressing less aggressive NPC mutations.

More work will be needed to establish unambiguously what is the mode of action of thioperamide, as well as the link between LBPA and cholesterol. However, some speculations are already possible. First, the notion that thioperamide acts by modulating the activity of the NPC1 or NPC2 proteins can be ruled out, given the effects of the drug in NPC cells and mice. It also seems fairly unlikely that thioperamide exerts its direct and selective effect on cholesterol, by modulating the biochemical or biophysical properties of the cholesterol-laden bilayer after partitioning into endosomal (LBPA-containing) membranes. Indeed, the drug raises LBPA levels specifically without affecting any other endosomal function or protein. Moreover, this view is not easily reconciled with observations that LBPA levels are also increased by other inhibitors of the HRH3 receptor that do not share a common backbone with thioperamide. It should be noted that HRH3 and HRH4 are expressed in most tissues, including liver and brain, as revealed by an analysis using Genevestigator tools that combine thousands of microarray experiments (https://www.genevestigator.com/gv/index.jsp). Hence, the simplest interpretation is that binding of thioperamide to the HRH3 receptor on the cell surface as inverse agonists [24] increases LBPA levels by stimulating biosynthesis or inhibiting turnover—an issue not easily solved since the metabolic regulation of LBPA remains mysterious and the enzymes involved in biosynthesis and turnover are not known [11,12]. High LBPA levels in turn may contribute to fluidify or buffer cholesterol in endosomal membranes, thereby facilitating its mobilization and subsequent export to cellular destinations.

In NPC cells, before the addition of thioperamide, the levels of LBPA are already strongly elevated [6] (see Fig 6B), as part of the general expansion of late endosome volume, protein and lipid [21]. This enlargement of the endosomal system presumably reflects the attempt to compensate for the accumulation of storage materials. Eventually, however, the system collapses under the excess load, leading to a traffic jam and a breakdown of endosomal membrane dynamics [26,27]. Given its role in endosomal cholesterol transport [6,14,28], we propose that LBPA then becomes limiting, because its capacity to accommodate—or buffer—excess cholesterol in NPC endosomes is overwhelmed. Alternatively, LBPA may play a more direct role in cholesterol mobilization and transport. In any case, it is also tempting to speculate that the highly selective changes in LBPA and sLBPA acyl chain composition—shorter chains and reduced number of double bonds—observed in NPC mouse liver reflect some additional adjustment in membrane chemical and physical properties to better accommodate changes due to cholesterol accumulation [47–50].

At first sight, it may appear counter-intuitive that thioperamide acts in NPC cells and mice via LBPA, since the drug decreases LBPA levels in NPC cells after 72 h (Fig 5B) and in $Npc1^{-/-}$ mouse liver at least to some extent (Fig 6B). However, our model predicts that LBPA will only be transiently increased in NPC cells. High LBPA levels will alleviate cholesterol overload, which in turn will reduce storage and revert the cell response to the endosome overload, resulting in decreased levels of LBPA and other endosomal lipids and proteins. Indeed, as predicted, a significant increase in LBPA levels was transiently observed in NPC cells at a shorter time-point, before changes in cholesterol could be detected (Fig 5B at 48 h). In NPC mice, the effects of the drug are also likely to be masked by the beneficial decrease in storage of cholesterol and other lipids including LBPA. Consistent with previous findings [28], we conclude that in NPC cells much like in other cell types, thioperamide increases LBPA levels, which helps buffer membrane cholesterol and reduce storage materials in endosomes. Since a very limited number of strategies only are emerging as possible treatment for NPC [31,43,51,52], we believe that our study of the FDA-approved compound thioperamide provides exciting novel perspectives for the treatment of NPC.

## Materials and Methods

### Reagents and cells

We used the following mouse monoclonal antibodies against transferrin receptor (13-6800, Invitrogen, 1/200); CD63 [53]; LBPA (1/100) [7]; and EGF receptor (555996, BD Biosciences, 1/200). We also used rabbit monoclonal antibodies against Lamp1 (D2D11, 9091, Cell Signaling, 1/500); and rabbit polyclonal antibody against EEA1 (ALX-210-239, ENZO life science, 1/400); as well as affinity-purified donkey antibodies against mouse IgG, Alexa 488-labelled (715,545,151, Jackson ImmunoResearch, 1/200) and against rabbit IgG, Cy3-labelled (ref: 711,165,152, Jackson ImmunoResearch, 1/200). We obtained filipin (F4767) from Sigma; BODIPY-493/503 Methyl Bromide (used at 1/1,000 from 1 mg/ml ethanol stock) and CellTracker green (C2925) from Thermo Fisher Scientific; and RNase (12091-021) and propidium iodide from (Ref. P3566) from Invitrogen.

The three NPC fibroblast patient cell lines were obtained from Coriell Institute, NIGMS Human Genetic Cell Repository (reference

numbers are indicated in the text and figures). All details about the NPC gene mutations can be found on the Coriell Institute website (https://www.coriell.org/1/NIGMS). When indicated, the cells were treated with 10 μM thioperamide or DMSO (1/2000) for 72 h and imaged by automated microscope.

### HCS screen

The compound screen was carried out in Hela MZ cells (provided by Marino Zerial, MPI-CBG, Dresden) and A431 cells, cultured as described [14]. HeLa and A431 cells are not on the list of commonly misidentified cell lines maintained by the International Cell Line Authentication Committee. Our HeLa MZ and A431 cells were authenticated by Microsynth (Balgach, Switzerland) and are mycoplasma-negative as tested by GATC Biotech (Konstanz, Germany). In the screen, we used BD Falcon 384-well imaging plates (ref. 353962), in duplicate for each cell line. In each well, cells in a 20 μl suspension (3,500 cells/well) were seeded on top of 5 μl solution containing 10 μM compound. In each plate, DMSO controls were added in all wells of column 2 and U18666A in all wells of column 23. Plates were incubated for 18 h before fixation (4% PFA) and staining using an automated plate washer (BioTek EL406). Cells were then incubated for 30 min with CellTracker green and stained as follows: Step 1: 6C4 monoclonal antibody against LBPA, 1/100, saponin 0.05%, RNAse 1/100, BSA 1%, in PBS, incubation for 1 h and Step 2: propidium iodide, filipin 1/50, Cy5-labelled secondary antibody against mouse IgG, 1/200, incubation for 30 min. Image acquisition was performed immediately after staining on a BD pathway 455 automated microscope with the 20× objective. Four images were captured per well.

### Image analysis with MetaXpress custom module editor

To analyse and quantify the LBPA and free cholesterol cell content, we used the MetaXpress Custom Module editor software to first segment the image and generate relevant masks, which were then applied on the fluorescent images to extract relevant measurements. In the first step, the cell was segmented by using the nuclei (propidium iodide) channel and the cytoplasm (CellTracker green) channel. To facilitate segmentation of LBPA endosome, we applied the top hat deconvolution method to the LBPA channel (Cy5) to reduce the background noise and highlight bright granules. LBPA endosomes were segmented from the deconvoluted image. When indicated, we used a perinuclear mask (4-μm-diameter region immediately outside the nucleus) to specifically quantify the LBPA and cholesterol (filipin) signals in the perinuclear area. The final masks are then applied to the original fluorescent images, and 50 measurements per cell (e.g. size, area, integrated intensity, average intensity and object count) are extracted. The same analysis pipeline has been applied to all images.

### Data analysis with AcuityXpress

All plates were annotated with the compound library using AcuityXpress. Dataset was normalized (Z-score) and analysed using the same software to generate all screen graphs and PCA. Compounds displaying more than 20% toxicity were excluded from the data analysis.

### EGF receptor degradation

HeLa MZ cells were starved in serum-free medium for 4 h and treated with 100 nM EGF as indicated [54]. Cells were then fixed and stained with antibodies to the EGF receptor, followed by fluorescently labelled secondary antibodies. The EGF receptor signal was then quantified by automated microscopy using the ImageXpress® confocal microscope (Molecular Devices™) after segmentation of the granular structures containing the EGFR signal [55], using Custom Module Editor™ from MetaXpress™. The obtained mask was then applied to the EGF receptor fluorescent micrographs, and the integrated intensity (sum of all pixel intensities) is extracted for each cell (approx. 5,000 cells analysed per condition).

### HRH3 stable cell

The HeLa MZ HRH3-GFP stable cell line has been generated using the gateway pLenti 6.3 V5CT vector backbone. Lentivirus has been prepared as follow: $6 \times 10^3$ HEK293T cells were seeded into 10-cm dishes pre-coated for 15 min with 0.001% poly-L-lysine (1:10 dilution of 0.01% stock). Cells were incubated for 6 h at 37°C, 5% $CO_2$ in medium containing 10% FBS before transfection. The transfection mix contained 15 μl Lipofectamine 2000, plasmids (pLenti 6.3-V5CT-HRH3-GFP 15 μg, pLP1 10 μg, pLP2 10 μg, pLP-VSV-G 10 μg) in Opti-MEM (500 μl final). The reactions were mixed and incubated for 20 min at room temperature. After 48 h, the medium was aspirated and exchanged with fresh medium. Then, 24 h later, the medium was collected, filtered (0.45 μm) and directly added onto target cells (at 40% confluency). The target cells were then cultured for 48 h before detecting expression. At that point, the medium was changed and cells treated with blasticidin (20 μg/ml) for the first selection. Clone selection was carried out in the presence of 5 μg/ml blasticidin by dilution in 384-well plates, and clones were analysed by automated microscope.

### CRISPR/Cas9 NPC1 and NPC2 KO cell lines

Guide sequences to produce NPC1 and NPC2 KO cells were obtained using the CRISPR design tool: NPC1: (fwd: CACCGCCAAGGGC CAGGCCGCGAG; rev: AAACCTCGCGGCCTGGCCCTTGGCG); NPC2: (fwd: TAATACGACTCACTATAGGTCCTTGAACTGCACC; rev: TTCT AGCTCTAAAACAACCGGTGCAGTTCAAGGA). The sequences were used to insert the target sequence into the pX330 vector using Golden Gate Assembly (New England Biolabs) and transfected into cells. Knockout clones were isolated by serial dilution and confirmed by RT–PCR, Western blotting and filipin staining.

### RT–PCR

Total RNA was extracted using RNeasy Mini Kit from Qiagen (ref. 74104) from monolayers of Hela according to manufacturer's recommendation. cDNA synthesis was carried out using SuperScript™ VILO™ cDNA Synthesis Kit (Life Technologies AG; Basel, Switzerland) from 250 ng of total RNA. mRNA expression was evaluated using SsoAdvanced SYBR Green Supermix (Bio-Rad Laboratories, Hercules, CA) with 10 ng of cDNA with specific primers of interest on a CFX Connect Real-time PCR Detection System (Bio-Rad). Relative amounts of mRNA were calculated by comparative CT analysis

with 18S ribosomal RNA used as internal control. All primers are QuantiTect primer from Qiagen (Hilden, Germany). Primers are as follows: ANXA1 (QT00078197); FYCO1 (QT00037009); OSBPL1A (QT00049196); STARD3 (QT00013517); VAPA (QT00033817); VAPB (QT02450693); RRN18S (QT00199367); LDLR_1 (QT00045864); and HMGCR (QT00004081).

## Lipid extraction for mass spec

Cells (near confluency in 6-cm plastic dishes) were scraped off the dish in PBS and sedimented for analysis. Alternatively, lyophilized homogenized liver tissue (7.5 mg) was resuspended in 100 μl cold water before addition of 360 μl methanol and the internal standard ergosterol (20 nmol). Next, 1.2 ml 2-methoxy-2-methylpropane (MTBE) was added and lipids were extracted by vortexing at 4°C for 10 min followed by 1-h shaking at room temperature to allow complete lipid partitioning [56]. A total of 200 μl water was added to induce phase separation, and the upper phase was collected and dried. Total phosphates were quantified with an ammonium molybdate colorimetric assay [57].

## Mass spectrometry

Membrane cholesterol and cholesteryl ester amounts were normalized and calibrated using the total phosphate content and the integrated signal of a spiked ergosterol standard, as previously described [14]. For phospholipid analysis, dried lipid samples were re-dissolved in chloroform–methanol (1:1 v/v). Separation was performed on a HILIC Kinetex Column (2.6 μm, $2.1 \times 50$ mm$^2$) on a Shimadzu Prominence UFPLC xr system (Tokyo, Japan): mobile phase A was acetonitrile:methanol 10:1 (v/v) containing 10 mM ammonium formate and 0.5% formic acid; mobile phase B was deionized water containing 10 mM ammonium formate and 0.5% formic acid. The elution of the gradient began with 5%B at a 200 μl/min flow and increased linearly to 50% B over 7 min, then the elution continued at 50% B for 1.5 min and finally the column was re-equilibrated for 2.5 min. The sample was injected in 2 μl chloroform:methanol 1:2 (v/v). Data were acquired in full-scan mode at high resolution on a hybrid Orbitrap Elite (Thermo Fisher Scientific, Bremen, Germany). The system was operated at 240,000 resolution ($m/z$ 400) with an AGC set at 1.0E6 and one microscan set at 10-ms maximum injection time. The heated electrospray source HESI II was operated in positive mode at a temperature of 90°C and a source voltage at 4.0KV. Sheath gas and auxiliary gas were set at 20 and 5 arbitrary units, respectively, while the transfer capillary temperature was set to 275°C. The mass spectrometry data were acquired with LTQ Tuneplus2.7SP2 and treated with Xcalibur 4.0QF2 (Thermo Fisher Scientific). Lipid identification was carried out with Lipid Data Analyzer II (LDA v. 2.5.2, IGB-TUG Graz University) [58]. Peaks were identified by their respective retention time, m/z and intensity. Instruments were calibrated to ensure a mass accuracy lower than 3 ppm. Data visualization was improved with the LCMSexplorer web tool hosted at EPFL (https://gecftools.epfl.ch/lcmsexplorer). We could clearly show the separation of LBPA from the isobaric lipid PG, as already described by others with HILIC column [59], which allows unambiguous quantification of the two lipids (Appendix Fig S1).

## Enzymatic cholesterol quantification

Cholesterol content in liver samples was quantified enzymatically using the Amplex Red Cholesterol Assay Kit (Molecular probes; A12216). Liver lipids extracted with MTBE were dried, resuspended in pure methanol and then directly diluted in the assay buffer. The assay in a 96-well plate format was according to the manufacturer's instructions.

## Mouse lipid analysis

BALBc/NPC$^{nih}$ mice were bred as heterozygotes to generate $Npc1^{-/-}$ mice and control genotypes. Mice were bred and housed under non-sterile conditions, with food and water available ad lib. All experiments were conducted using protocols approved by the UK Home Office Animal Scientific Procedures Act, 1986. $Npc1^{-/-}$ and $Npc1^{+/+}$ mice ($n = 6$ males and 6 females) were treated with the compound in the drinking water (at 8.8 mg/l, corresponding to 2 mg/kg/day). Treatment started at weaning (3 weeks of age), and mice were sacrificed at 9 weeks of age by intraperitoneal injection with an overdose of phenobarbital. Care was taken to minimize suffering through euthanizing the animals once liver enlargement was unambiguously detected by visual inspection (i.e. 3 weeks after initiation of treatment).

## Mouse behavioural analysis

Thioperamide was delivered as above, and miglustat (600 mg/kg) was administered as a dry admix in powdered chow from 3 weeks of age. The weight and spontaneous activity of each mouse were recorded weekly [60] until they reached the humane end-point (loss of 10% body weight within 24 h). Motor function and coordination were assessed by observational counting of total rearing events recorded manually for 5 min (the number of times the mouse reared on its hind legs with or without support of the cage wall). Tremor was measured [60] using a commercial tremor monitor (San Diego Instruments) according to manufacturer's instructions on an anti-vibration table. The mice were monitored for 256 s, after a 30-s acclimatization period. The tremor monitor was connected to a computer via National instrument PCI card, and the output (amplitude/time) was analysed using LabView software to give a measurement of power for each frequency (0–64 Hz).

## Other methods

Transfection with siRNAs was performed according to manufacturers' instructions using RNAiMAX (Thermo Fisher Scientific). We performed reverse transfection by seeding 80 μl HeLa cells ($2 \times 10^3$ cells/well) onto a 20 μl mix of siRNA (10 nM) and transfection reagent. Hela MZ cells were infected with VSV [19] using a BioTek™ EL406 plate washer, fixed with 4% PFA and stained with DAPI, and images were acquired with the IXM™ microscope and analysed with MetaXpress™ to quantify infected cells. The cholesterol content of liver extracts was measured enzymatically using the Amplex Red Kit (Molecular Probes). Dried lipid samples were re-dissolved in pure methanol, sonicated for 5 min and diluted in the assay buffer for quantification. Electron microscopy after plastic embedding for HRP analysis [61] or after immunogold labelling of cryosections [62] has

been described. Quantitation of LBPA labelling of sections from control (DMSO-treated) or thioperamide-treated cells was performed in a double-blind fashion by counting the number of gold particles per endosomes in each endosome identified in two sets of 16 micrographs for each condition (DMSO- and thioperamide-treated cells).

**Expanded View** for this article is available online.

## Acknowledgements

The authors acknowledge the use of the Australian Microscopy & Microanalysis Research Facility at the Center for Microscopy and Microanalysis at The University of Queensland. Support was from the Swiss National Science Foundation, the NCCR in Chemical Biology and LipidX from the Swiss SystemsX.ch Initiative, evaluated by the Swiss National Science Foundation (to J. G.), from the European Union Seventh Framework Programme (FP7 2007—2013) under grant agreement no. 289278—"Sphingonet" (AC/FP) (to F.M.P.), from grants and a fellowship from the National Health and Medical Research Council of Australia (grant numbers APP1037320, APP1058565 and APP569542 (to RGP)). F.M.P. is a Royal Society Wolfson Research Merit Award holder and a Wellcome Trust Investigator in Science. FV has been supported by a fellowship from the National Niemann-Pick C Disease Foundation (NNPDF).

## Author contributions

DM designed the experiments, carried out the screen and the characterization as well as the data analysis, and wrote the manuscript. FV contributed to the lipid analysis. SV contributed to the screen. CS contributed to data analysis. AC carried out the experiments with mice. JPM contributed to the mass spectrometry analysis. CF contributed to the electron microscopy analysis. MD contributed to the analysis of the mice. MM contributed to the mass spectrometry analysis. RGP contributed to the electron microscopy analysis. FMP contributed to the experiments with mice. JG designed the experiments and wrote the manuscript.

## Conflict of interest

The authors declare that they have no conflict of interest.

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
