## [Review Process File · EMBO Reports]

Drug-induced increase in lysobisphosphatidic acid reduces the cholesterol overload in Niemann-Pick type C cells and mice

Dimitri Moreau, Fabrizio Vacca, Stefania Vossio, Cameron Scott, Alexandria Colaco, Jonathan Paz Montoya, Charles Ferguson, Markus Damme, Marc Moniatte, Robert G. Parton, Frances M. Platt and Jean Gruenberg

Review timeline:

Submission date:	13 September 2018
Editorial Decision:	24 October 2018
Revision received:	7 March 2019
Editorial Decision:	5 April 2019
Revision received:	12 April 2019
Accepted:	23 April 2019

Transaction Report:

1st Editorial Decision

24 October 2018

Thank you for the submission of your research manuscript to our journal. We have now received the full set of referee reports that is copied below.

As you will see the referees acknowledge the potential interest of your findings but they all point out a number of limitations that I briefly summarize below:

- 1) Limited information on the screen provided
- 2) Missing information on the cytotoxicity of thioperamide
- 3) Reproducibility and missing quantification/statistics => how many independent experiments have been performed?
- 4) The effect of thioperamide on cholesterol storage in NPC fibroblasts are not fully convincing
- 5) Biochemical evidence on lipid mobilization is missing
- 6) Insufficient clinical evaluation, more pathological parameters should be assessed in the mouse.
- 7) Mechanistic insight is limited => how does thioperamide act on LBPA?

I have discussed these points further with the referees and they all emphasized that it will be essential to address points 1-6 to substantiate the findings and strengthen the study. In particular, it will be important to reinforce the translational value of your findings (point 6). Referee 2 suggested to provide at least a minimal survival experiment and to determine lipid levels in the brain after treatment.

However, all referees agreed that further mechanistic insight is not necessary (point 7 above, referee 3) and can be saved for a future study and/or the discussion.

Given these constructive comments and your feedback that you will be able to address these concerns in a reasonable timeframe, we would like to invite you to revise your manuscript with the

understanding that the referee concerns (as detailed above and in their reports) must be fully addressed and their suggestions taken on board. Please address all referee concerns in a complete point-by-point response. Acceptance of the manuscript will depend on a positive outcome of a second round of review. It is EMBO reports policy to allow a single round of revision only and acceptance or rejection of the manuscript will therefore depend on the completeness of your responses included in the next, final version of the manuscript.

Revised manuscripts should be submitted within three months of a request for revision; they will otherwise be treated as new submissions. Please contact us if a 3-months time frame is not sufficient for the revisions so that we can discuss the revisions further.

Supplementary/additional data: The Expanded View format, which will be displayed in the main HTML of the paper in a collapsible format, has replaced the Supplementary information. You can submit up to 5 images as Expanded View. Please follow the nomenclature Figure EV1, Figure EV2 etc. The figure legend for these should be included in the main manuscript document file in a section called Expanded View Figure Legends after the main Figure Legends section. Additional Supplementary material should be supplied as a single pdf labeled Appendix. The Appendix includes a table of content on the first page with page numbers, all figures and their legends. Please follow the nomenclature Appendix Figure Sx throughout the text and also label the figures according to this nomenclature. For more details please refer to our guide to authors.

Regarding data quantification, please ensure to specify the name of the statistical test used to generate error bars and P values, the number (n) of independent experiments underlying each data point (not replicate measures of one sample), and the test used to calculate p-values in each figure legend. Discussion of statistical methodology can be reported in the materials and methods section, but figure legends should contain a basic description of n, P and the test applied. Please also include scale bars in all microscopy images.

We now strongly encourage the publication of original source data with the aim of making primary data more accessible and transparent to the reader. The source data will be published in a separate source data file online along with the accepted manuscript and will be linked to the relevant figure. If you would like to use this opportunity, please submit the source data (for example scans of entire gels or blots, data points of graphs in an excel sheet, additional images, etc.) of your key experiments together with the revised manuscript. Please include size markers for scans of entire gels, label the scans with figure and panel number, and send one PDF file per figure.

- a complete author checklist, which you can download from our author guidelines (<http://embor.embopress.org/authorguide#revision>). Please insert page numbers in the checklist to indicate where the requested information can be found.
 - a letter detailing your responses to the referee comments in Word format (.doc)
 - a Microsoft Word file (.doc) of the revised manuscript text
 - editable TIFF or EPS-formatted figure files in high resolution
- (In order to avoid delays later in the publication process please check our figure guidelines before preparing the figures for your manuscript:
http://www.embopress.org/sites/default/files/EMBOPress_Figure_Guidelines_061115.pdf)
- a separate PDF file of any Supplementary information (in its final format)
 - all corresponding authors are required to provide an ORCID ID for their name. Please find instructions on how to link your ORCID ID to your account in our manuscript tracking system in our Author guidelines (<http://embor.embopress.org/authorguide>).

As part of the EMBO publication's Transparent Editorial Process, EMBO reports publishes online a Review Process File to accompany accepted manuscripts. This File will be published in conjunction

with your paper and will include the referee reports, your point-by-point response and all pertinent correspondence relating to the manuscript.

I look forward to seeing a revised version of your manuscript when it is ready. Please let me know if you have questions or comments regarding the revision.

REFeree REPORTS

Referee #1:

Moreau et al report the results of a high content, image based screen designed to identify small molecules that increase LBPA without concurrently increasing cellular cholesterol. The rationale for the screen is that LBPA plays an important role in mobilizing endolysosomal cholesterol, thereby providing a novel approach to alleviate the storage of unesterified cholesterol in Niemann-Pick type C disease (NPC). The screen was performed using the Prestwick library of FDA approved drugs. The data presented suggest that the compound identified in the screen, thioperamide, has desired activity. However, several notable limitations are present in the data as it is currently presented.

Limited information on the screen is provided. What is the z-factor for the filipin and LBPA readout? What criteria were used to identify hits?

Multipoint (8 or 11pt) dose-response curves with thioperamide and pitolisant, as well as a time course of the effect, should be included.

Quantification of cytotoxicity for thioperamide and pitolisant following treatment of HeLa cells and NPC patient fibroblasts (at effective concentrations/durations) should be included.

In almost all of the figures, statistical analysis to assess significance is missing. Additionally, none of the figure legends provide an indication of the number of experimental replicates used for quantification. This latter point is concerning for many figures, including 2C, 3B-D. Together, these deficits raise concerns about the rigor of the analyses.

Fig 3A: specify criteria used to identify late endosomes.

Fig 4: The authors state no change was observed for "distribution or amounts" of the markers studied, but quantitative data are only shown for amounts. Similar quantification should be shown for distribution or the statement in the text should be modified.

Fig 4 lacks a positive control such as treatment with U18666A.

Fig 5B should include quantification with appropriate statistics.

The experiments in Fig 5C-G are unclear. It appears that these cells are not treated with thioperamide, raising the question: Dose knockdown of endogenous HRH3-4 on HeLa cells abrogate the response to drug?

Fig 6 shows effects on cholesterol storage of NPC fibroblasts at 72 hrs, but these results are complicated by the fact that LBPA levels are decreased at this time point, not increased. The authors suggest this may reflect a transient increase in LBPA at 48 hr, but are these differences significant? Alternatively, is the dose/duration of treatment toxic to fibroblasts? Does it alter fibroblast density, a factor known to influence filipin positivity in vitro? Is the beneficial effect dependent upon expression of HRH3-4?

The authors state that Fig 6C shows thioperamide efficiently reduced cholesterol levels in 3 NPC lines, but these changes are significant in only 2 lines. More careful wording would be appropriate.

If thioperamide is working to mobilize endolysosomal cholesterol in NPC cells, as the authors suggest, is there biochemical evidence from lipid analysis or gene expression studies to support this conclusion?

It would be appropriate to include data summarized in Fig 5a as a supplement.

Was toxicity noted following treatment of mice with thioperamide as reflected by changes in liver enzymes, renal function or body weight?

In the Discussion, it would be appropriate to mention whether hepatocytes and neurons are known to express HRH3-4?

Referee #2:

Previous work from these authors showed that LBPA mediates cholesterol flux through endosomes and that interfering with LBPA function causes endosomal cholesterol accumulation phenocopying Niemann Pick type C disease (NPC). Based on these findings the authors screen here for compounds that, by influencing LBPA levels or distribution, could counteract cholesterol accumulation. They identify thioperamide as a novel modulator of LBPA levels in late endosomes that decreases cholesterol overload in fibroblasts from NPC patients and in the liver of NPC ko mice. Finding therapies for a fatal disease like NPC is indeed of interest and the results here presented have translational value. However, there are some important concerns:

-About the novelty and significance. The screening method is not new and was used by the authors in a previous study where they identified Wnt pathway as a regulator of intracellular cholesterol transport (Scott et al., EMBO Rep 2015). Same screening method is used here to find modulators of the known cholesterol efflux effector LBPA. While the drug identified, thioperamide, has the potential of becoming a novel therapeutic strategy for NPC, the molecular mechanisms underlying its mode of action on LBPA remain undetermined despite the author efforts. While correlative evidence suggest the involvement of histamine receptors their role is not clear as well as the means by which LBPA facilitates endosomal cholesterol efflux.

-About the preclinical validation. Analysis of the effects of thioperamide treatment in the NPC mouse model is restricted to determine LBPA and cholesterol levels in the liver. This limited analysis weakens the translational potential of the study. Does this drug impact other pathological parameters in liver tissue (i.e. lysosomal size, inflammation) Does it impact the brain, which is particularly affected in NPC? Does it extend life span?

-About the data presentation. Results should be more carefully illustrated in the figures. I summarize below instances that need revision:

Statistical significance is missing in many graphs (i.e. Figures 1D, 2B, 3B, 4C, 6B, 6D, 6F)

Figure 1B. The plot is confusing and needs explanation. Why DMSO and U18666A data appear as a group of dots while thioperamide and trimeprazine treated are shown just like a square?

Figure 1E. Accurate quantification is missing. Bar length is not indicated in the figure legend

Figure 2B. Why is Pitolisant used here instead of trimeprazine, which was used in Figures 1 and 2C?

Clarity of the manuscript would benefit from more self-explanatory subtitles in the main text and in the figure legends

Referee #3:

The manuscript by Moreau et al screened the FDA-approved drug library for the chemicals to ameliorate cholesterol accumulation in lysosome in NPC cells. They found that thioperamide maleate (Thio) was able to eliminate lysosomal cholesterol accumulation by elevating the LBPA level. Thio seemed not impair endosomal functions. They further applied Thio to fibroblasts from NPC patients and NPC1-deficient mouse. The results consistently showed that Thio reduced cholesterol accumulation. The current study is well-conducted and the data appear solid.

Comments:

1) The mechanism of LBPA and Thio-mediated cholesterol efflux from lysosomes should be further investigated. It has been known that membrane contact sites and sterol transfer proteins play essential role in intracellular cholesterol traffic. The authors may investigate whether LBPA or Thio alter membrane contacts of lysosome-ER, lysosome-peroxisome, et al. Are the STPs changed in Thio treated cells?

2) It is very important for the authors to show that Thio has effect on human patients' cells and NPC disease model. The authors need to further analyze whether Thio improve neuron-muscle function of the NPC1^{-/-} mouse or prolong the life-span of the mice.

1st Revision - authors' response

7 March 2019

Referee #1:

Moreau et al report the results of a high content, image-based screen designed to identify small molecules that increase LBPA without concurrently increasing cellular cholesterol. The rationale for the screen is that LBPA plays an important role in mobilizing endolysosomal cholesterol, thereby providing a novel approach to alleviate the storage of unesterified cholesterol in Niemann-Pick type C disease (NPC). The screen was performed using the Prestwick library of FDA approved drugs. The data presented suggest that the compound identified in the screen, thioperamide, has desired activity. However, several notable limitations are present in the data as it is currently presented.

We thank the reviewer for his/her assessment and we have addressed below the limitations outlined by this reviewer.

Limited information on the screen is provided. What is the z-factor for the filipin and LBPA readout? What criteria were used to identify hits?

We are sorry if information on the screen was missing, partly because of the space constraints of the journal. We wish to emphasize the fact that our goal was not to identify “hits” per se. The previous work of us and others showed that experimental conditions that increase the cholesterol content of endolysosomes, including drugs (e.g. U18666A) or mutations (e.g. NPC1), also increase the LBPA content. Our original goal was thus to identify compounds that modify LBPA but not cholesterol. Using the Prestwick library, we found that thioperamide increased LBPA 3X without affecting cholesterol (Fig 1D) - not surprisingly, some Prestwick compounds showed a U18666A-like behavior (e.g. trimeprazine) — and we decided to further characterize thioperamide. We have now clarified these issues and added the z-factor for the LBPA and filipin readout.

Multipoint (8 or 11pt) dose-response curves with thioperamide and pitolisant, as well as a time course of the effect, should be included.

As requested, we have added the dose-response curve with thioperamide and pitolisant, as well as a time-courses of the effects. These data show that the dose-response profiles (new Fig EV3A) and timecourses (new Fig EV3B) of LBPA accumulation are very similar. However, these data also show that pitolisant may have some cytotoxic effects after longer times (new Fig EV3C), confirming our decision to center our efforts on thioperamide.

Quantification of cytotoxicity for thioperamide and pitolisant following treatment of HeLa cells and NPC patient fibroblasts (at effective concentrations/durations) should be included.

We apologize if this point was not clear. Previous studies have established that thioperamide is not toxic in animal models (e.g. ref 25). Consistent with these data, thioperamide showed not toxicity in our screen. We have now included new data on the effects of thioperamide in HeLa cells (new Fig EV3) and in the NPC1 and NPC2 cell-lines that we used (new Fig 5D-E). These data show that thioperamide is not toxic in HeLa and NPC cells. In fact, thioperamide somewhat increased cell

survival in NPC cells. This has been clarified in the new version of the paper.

In almost all of the figures, statistical analysis to assess significance is missing. Additionally, none of the figure legends provide an indication of the number of experimental replicates used for quantification. This latter point is concerning for many figures, including 2C, 3B-D. Together, these deficits raise concerns about the rigor of the analyses.

We apologize if the statistical was missing. As requested this information was added, as well as the number of experimental replicates in all Figures. In Fig 2C and 3C-D, the data from the screen are plotted and thus each dot corresponds to the average of replicates in the duplicate plates (as in Fig 1B), and ≥ 600 cells were analysed per compound (only compounds that showed $< 20\%$ toxicity are plotted). In the EM analysis after immuno-gold labelling of cryosections, Fig 3B represents the double-blind quantification of the mean of two replicates for each condition. This was not clarified in the Legends of the Figures.

Fig 3A: specify criteria used to identify late endosomes.

Fig 3A shows cryo-sections labeled with an anti-LBPA antibody. We and others have found that LBPA is only present in late endosomes (e.g. ref 6 and 7 by Kobayashi et al.). Thus, LBPA is used to identify late endosomes in Fig 3A. This was clarified in the Legend of Fig 3.

Fig 4: The authors state no change was observed for "distribution or amounts" of the markers studied, but quantitative data are only shown for amounts. Similar quantification should be shown for distribution or the statement in the text should be modified.

As requested, we now show that the distribution of the markers is not affected in the new Fig EV2.

Fig 4 lacks a positive control such as treatment with U18666A.

As requested, we now show the positive control with U18666A in the new Fig EV2.

Fig 5B should include quantification with appropriate statistics.

The effects of pitolisant on LBPA accumulation has been quantified in the new version of the paper, including the dose-response and the time-course of the effects of the drug (new Fig EV3A and EV3B).

The experiments in Fig 5C-G are unclear. It appears that these cells are not treated with thioperamide, raising the question: Dose knockdown of endogenous HRH3-4 on HeLa cells abrogate the response to drug?

We apologize if this was not clear in the original version. Thioperamide is believed to act as an inverse agonist of HRH3, i.e. it decreases the constitutive or intrinsic activity of the receptor in the absence of ligand. Consistent with this notion, we find that the levels of LBPA and HRH3 are inversely-correlated. Indeed, in a mixed population of cells expressing GFP-tagged HRH3 (ex-Fig 5F, new Fig 4F), the cellular intensity of the LBPA signal is skewed towards cells expressing low levels of HRH3-GFP and vice-versa (new Fig 4G), indicating that LBPA levels are low in cells expressing high HRH3-GFP levels and high in cells expressing low HRH3-GFP levels. We confirmed these observations using cells stably expressing HRH3-GFP. In these cells, HRH3-GFP knockdown (new Fig 4C and E) was accompanied with a concomitant increase in LBPA levels (new Fig 4D-E) – further demonstrating that LBPA and HRH3 levels are anti-correlated. This was now clarified in the new version of the manuscript.

Fig 6 shows effects on cholesterol storage of NPC fibroblasts at 72 hrs, but these results are complicated by the fact that LBPA levels are decreased at this time point, not increased. The authors suggest this may reflect a transient increase in LBPA at 48 hr, but are these differences significant? Alternatively, is the dose/duration of treatment toxic to fibroblasts? Does it alter fibroblast density, a factor known to influence filipin positivity in vitro? Is the beneficial effect dependent upon expression of HRH3-4?

We thank the reviewer for this comment. After 48h thioperamide treatment of NPC fibroblasts, the increase in LBPA levels is highly significant, close to 2X when compared to DMSO controls (ex-Fig 6B, new Fig 5B). In fact, the effects of the drug are particularly striking, since even before thioperamide addition, LBPA levels are already much higher in NPC cells when compared to controls (e.g. see our quantification of LBPA in NPC mice, new Fig 6B). This issue has been clarified in the text. In addition, the decrease in LBPA levels after 72h is clearly not due to some toxic effect of the compound. In fact, our new data (new Fig 5E) show that thioperamide increases

cell survival in the NPC1 and NPC2 cell lines.

We could not determine whether the beneficial effects of thioperamide depends on the expression of HRH3-4. Indeed, we were unable to transfect NPC1 or NPC2 fibroblasts with siRNAs. More important, and as noted above, thioperamide acts as an inverse agonist, and thus HRH3-4 KD has the same effect as thioperamide on LBPA levels (new Fig 4F-G), making it impossible to discriminate between the effects of the drug and of the knockdown.

The authors state that Fig 6C shows thioperamide efficiently reduced cholesterol levels in 3 NPC lines, but these changes are significant in only 2 lines. More careful wording would be appropriate.

We apologize if the text was not clear. The effects of thioperamide on cholesterol accumulation in all 3 NPC cell lines are actually very strong. Thioperamide caused a highly significant (approx. 3X) decrease in endosomal cholesterol levels visualized with filipin (ex-Fig 6A, new Fig 5A, quantification in Fig 5B). Ex-Fig 6C (new Fig 5C) illustrates the effects of the drug on total cellular cholesterol, analyzed by mass spectrometry. The Fig shows that total cellular cholesterol is also reduced, but to a lesser extent: the effects of the drug appear less prominent than at the endosomal level (Fig 5B), because in NPC fibroblasts, the amounts of cholesterol accumulated in endosomes corresponds only to a fraction of total cellular cholesterol. This issue was clarified in the revised manuscript. It should also be noted that thioperamide reduced total cellular cholesterol levels as efficiently as cyclodextrin (Fig 6C), which is considered to be one of the — if not the — most efficient protocol to reduce cholesterol levels in cultured cells.

If thioperamide is working to mobilize endolysosomal cholesterol in NPC cells, as the authors suggest, is there biochemical evidence from lipid analysis or gene expression studies to support this conclusion?

We thank the reviewer for this comment. As requested, we have tested whether endolysosomal cholesterol is mobilized after thioperamide treatment. To this end, we generated NPC1 and NPC2 KO cells using CRISP/Cas9. Our new data show that thioperamide was able to partially correct the defect in transcriptional regulation of two canonical cholesterol-dependent genes, the LDL receptor and HMG CoA reductase, in cells NPC1 or NPC2 KO cells (new Fig 6A).

It would be appropriate to include data summarized in Fig 5a as a supplement.

As requested, we have added a supplementary table (Table EV3) that contains the data of ex Fig 5a – new Fig 4a.

Was toxicity noted following treatment of mice with thioperamide as reflected by changes in liver enzymes, renal function or body weight?

No toxicity of thioperamide was observed in mice. The data are included in the new Fig EV9, containing the analysis of mice treated with thioperamide (in response to the comments of reviewer 3).

In the Discussion, it would be appropriate to mention whether hepatocytes and neurons are known to express HRH3-4?

We thank the reviewer for this comment. As requested, we now mention in the Discussion that HRH3-4 is expressed in hepatocytes and in neurons. Indeed, an analysis using Genevestigator tools that combine thousands of microarray experiments (<https://www.genevestigator.com/gv/index.jsp>) shows that HRH3-4 are expressed in most tissues, including liver and brain. This issue has been clarified in the paper.

Referee #2:

Previous work from these authors showed that LBPA mediates cholesterol flux through endosomes and that interfering with LBPA function causes endosomal cholesterol accumulation phenocopying Niemann Pick type C disease (NPC). Based on these findings the authors screen here for compounds that, by influencing LBPA levels or distribution, could counteract cholesterol accumulation. They identify thioperamide as a novel modulator of LBPA levels in late endosomes that decreases cholesterol overload in fibroblasts from NPC patients and in the liver of NPC ko mice. Finding therapies for a fatal disease like NPC is indeed of interest and the results here presented have translational value. However, there are

some important concerns:

-About the novelty and significance. The screening method is not new and was used by the authors in a previous study where they identified Wnt pathway as a regulator of intracellular cholesterol transport (Scott et al., EMBO Rep 2015). Same screening method is used here to find modulators of the known cholesterol efflux effector LBPA. While the drug identified, thioperamide, has the potential of becoming a novel therapeutic strategy for NPC, the molecular mechanisms underlying its mode of action on LBPA remain undetermined despite the author efforts. While correlative evidence suggest the involvement of histamine receptors their role is not clear as well as the means by which LBPA facilitates endosomal cholesterol efflux.

This reviewer correctly points out that the mechanism of action of thioperamide is not clear yet, despite much effort. We agree that this issue is important but it extends beyond the present work. However, in response to the comments of referee 3, we have tested whether the expression of proteins involved in sterol transfer or membrane contact sites was changed after thioperamide treatment of control cells, NPC1 KO cells or NPC2 KO cells (we generated NPC1 and NPC2 KO cells using CRIPR/Cas9). We tested the following candidates (from the work of the indicated group): FYCO1 (H. Stenmark) ANXA1 (C. Futter), STARD3 (F. Alpy), and finally ORPIL, VAPA and VAPB (J. Neefjes), We did not see any significant change in the expression levels of any candidate protein, whether in controls or in NPC1 or NPC2 KO cells. These data are now shown in new Fig EV4.

-About the preclinical validation. Analysis of the effects of thioperamide treatment in the NPC mouse model is restricted to determine LBPA and cholesterol levels in the liver. This limited analysis weakens the translational potential of the study. Does this drug impact other pathological parameters in liver tissue (i.e. lysosomal size, inflammation) Does it impact the brain, which is particularly affected in NPC? Does it extend life span?

We thank the reviewer for this comment. As requested, we have now included new data in mice (New Fig EV9). In these studies, we analyzed the same mice that we had used in the paper. These mice lack NPC1 and thus model the most aggressive early onset form of the disease. Thioperamide did not significantly improve the life span, motor function/rearing or high frequency tremor, although some benefits were observed when combined with Miglustat (Fig EV9) — the only compound available as a treatment against NPC in Europe, but not in the US, which prolongs life but does not arrest disease progression. We have now included these new data in mice as new Fig EV9. Clearly, it will be interesting in the future to test the effects of thioperamide alone or in combination with Miglustat in mice expressing less aggressive NPC mutations.

-About the data presentation. Results should be more carefully illustrated in the figures. I summarize below instances that need revision:

As requested, data are more carefully presented in the figures.

Statistical significance is missing in many graphs (i.e. Figures 1D, 2B, 3B, 4C, 6B, 6D, 6F)
Statistical information was provided for the all figures, including Figures 1D, 2B, 3B, 4C, 6B, 6D, 6F.

Figure 1B. The plot is confusing and needs explanation. Why DMSO and U18666A data appear as a group of dots while thioperamide and trimeprazine treated are shown just like a square?

We are sorry if this was not clear. The plot shows the mean of duplicate values for each compound that have been tested in the screen, and therefore thioperamide and trimeprazine each appear as one square. In the screen, each 384-well plates included a full column of 12 wells treated with U18666A, as positive controls. and all these controls appear as the cluster of green dots in the plot. Similarly, each 384-well plates also included a full column of 12 wells treated with DMSO, as negative controls, and these cluster together close to the intercept between X- and Y-axis. In the original plot, the red dots unfortunately disappeared behind the black dots, and this has been changed in the new version of the plot. This issue was clarified in the new version of the paper.

Figure 1E. Accurate quantification is missing. Bar length is not indicated in the figure legend
We are sorry if this was not clear. The data are quantified precisely in Fig 1D, and this issue was clarified in the text. In the original version of the paper, the bar length was indicated on the Fig

itself. As requested, we have now added the bar length in the Legends.

Figure 2B. Why is Pitolisant used here instead of trimeprazine, which was used in Figures 1 and 2C?

We are sorry if this was not clear. We only used trimeprazine as an example of compounds that show U18666A-like effects in the screen itself, e.g. Fig 1B, D-E and Fig 2C. Pitolisant, however, is not in the Prestwick library and it was thus not tested in the screen. We found Pitolisant when searching for other compounds known to target the histamine receptors H3 (HRH3) and H4 (HRH4). We therefore used Pitolisant in Fig 2B because it targets HRH3-4 and because the effects of this compound are close to thioperamide, since it increased LBPA levels without affecting cholesterol. This has been clarified in the paper.

Clarity of the manuscript would benefit from more self-explanatory subtitles in the main text and in the figure legends

As requested, we have added self-explanatory subtitles in text and in legends.

Referee #3:

The manuscript by Moreau et al screened the FDA-approved drug library for the chemicals to ameliorate cholesterol accumulation in lysosome in NPC cells. They found that thioperamide maleate (Thio) was able to eliminate lysosomal cholesterol accumulation by elevating the LBPA level. Thio seemed not impair endosomal functions. They further applied Thio to fibroblasts from NPC patients and NPC1-deficient mouse. The results consistently showed that Thio reduced cholesterol accumulation. The current study is well-conducted and the data appear solid.

Comments:

1) The mechanism of LBPA and Thio-mediated cholesterol efflux from lysosomes should be further investigated. It has been known that membrane contact sites and sterol transfer proteins play essential role in intracellular cholesterol traffic. The authors may investigate whether LBPA or Thio alter membrane contacts of lysosome-ER, lysosome-peroxisome, et al. Are the STPs changed in Thio treated cells?

As discussed above (answers to the Editor and to referee 2), this reviewer correctly points out that the mechanism of action of thioperamide is not clear yet, despite much effort. We agree that this issue is important but it extends beyond the present work. However, we addressed the comment that membrane contact sites and sterol transfer proteins play a role in intracellular cholesterol traffic. As requested, we tested whether the expression of STPs or MCS proteins that play a role in cholesterol transfer at MCSs was changed after thioperamide treatment of control cells, NPC1 KO cells or NPC2 KO cells (we generated NPC1 and NPC2 KO cells using CRIPR/Cas9). We tested the following candidates (from the work of the indicated group): FYCO1 (H. Stenmark) ANXA1 (C. Futter), STARD3 (F. Alpy), and finally ORP1L, VAPA and VAPB (J. Neefjes). We did not see any significant change in the expression levels of any candidate in controls or in NPC1 or NPC2 KO cells. These data are now shown in new Fig EV4.

2) It is very important for the authors to show that Thio has effect on human patients' cells and NPC disease model. The authors need to further analyze whether Thio improve neuron-muscle function of the NPC1^{-/-} mouse or prolong the life-span of the mice.

We thank the reviewer for this comment. As requested, we have now included new behavioral data in mice (new Fig EV9). This analysis was carried out in the same mice that we had used in the paper. These mice lack NPC1 and thus model the most aggressive early onset form of the disease. Thioperamide did not significantly improve the life span, motor function/rearing or high frequency tremor, although some benefits were observed when combined with Miglustat (Fig EV9) — the only compound available as a treatment against NPC in Europe, but not in the US, which prolongs life but does not arrest disease progression. We have now included these new data in mice as new Fig EV9. Clearly, it will be interesting in the future to test the effects of thioperamide alone or in combination with Miglustat in mice expressing less aggressive NPC mutations.

Thank you for the submission of your revised manuscript to EMBO reports. It has been evaluated again by referee 1 and 2 and we have now received the full set of referee reports (copied below).

As you will see, both referees are very positive about the study and request only minor changes to clarify text, figures and data quantification.

From the editorial side, there are also a few things that we need before we can proceed with the official acceptance of your study.

- Please provide up to five keywords on the first page of the manuscript
- Author contributions: The contribution of Markus Damme is missing
- Please rename "Competing interests" paragraph to "Conflict of interest"
- Supplemental information: Please note that we can only accommodate up to five figures in the Expanded View Content. Please choose which ones you want to keep as EV content and provide these as individual files (one file per figure). The rest should go into an Appendix. The Appendix is a single pdf file, which includes a table of content on the first page with page numbers, all figures and their legends. Please follow the nomenclature Appendix Figure Sx throughout the text and also label the figures according to this nomenclature. For more details please refer to our guide to authors.
- We encourage authors to arrange the figure panels in a manner that they can be called out in the correct order in the text. This is currently not the case for Figure 1. Moreover, Fig 4 F,G are called out before Fig 4D and 5B is discussed after 5E. If possible the panels should be rearranged.
- Moreover, we noticed a callout to Fig 5 G (page 9), which is not present.
- Table EV1, EV2 and EV3 should be supplied as Datasets (Dataset EV1, Dataset EV2, Dataset EV3). Please remove the legend from the main manuscript file and provide it in the first row of the excel file.
- Movie EV1: Please provide the legend in a separate readme.txt file. Then zip Movie and legend together and upload this .zip file.
- Our data editors from Wiley have already inspected the Figure legends for completeness and accuracy. Please see their suggested changes in the attached Word file (and some more suggestions below). I have also taken the liberty to make a minor change to the Abstract. Moreover, the title may not exceed 100 characters. Could you please shorten it?
- Please provide scale bars for Figs 1E; Figs EV 2B,C,D. Please also define the size of all scale bars in the figure legends.
- Figure 4E, 5B: since the quantification is based on 2 replicates, which is not ideally suited to support a statistical analysis, I would suggest removing the p-values from these panels.
- Finally, EMBO reports papers are accompanied online by A) a short (1-2 sentences) summary of the findings and their significance, B) 2-3 bullet points highlighting key results and C) a synopsis image that is 550x200-400 pixels large (width x height). You can either show a model or key data in the synopsis image. Please note that the size is rather small and that text needs to be readable at the final size. Please send us this information along with the revised manuscript.

We look forward to seeing a final version of your manuscript as soon as possible. Please let me know if you have questions or comments regarding the revision.

REFEREE REPORTS

Referee #1:

This revised manuscript is much improved and the modifications address many of the concerns raised in the prior review. A few points remain:

Fig 3B: How many cells were used for quantification? Analysis of biological triplicates with assessment of statistical significance would be most appropriate. As the data are currently presented (mean values), there is no indication of variation between samples.

Fig EV9B, C: How many mice/group? Include error bars and statistical analysis to support conclusion in panel C that thioperamide plus miglustat is more beneficial than either treatment alone. Include thioperamide alone group.

Legend to EV9: Delete mention of rearing data, which is not included in figure.

Legend to Fig 1E: add scale bar size

Page 9: LBPA quantification in NPC1 mice is shown in Fig 6B rather than 5G

Referee #2:

The authors have addressed my queries and made an effort to further characterise thioperamide mode of action and the effects of the treatment in the mouse brain. Although, unfortunately, the effects in the brain and life span are not significant the underlying reasons for that are discussed. Overall, the results encourage further research on LBPA function and open perspectives on its use as a target for NPC treatment.

There are only two minor points that should be addressed but do not need re-review from my part.

1- Fix the apparent inconsistency on the following: While the subtitle of Figure 3 claims that Thioperamide affects endosome morphology, the text in page 5 states "analysis by electron microscopy showed that the ultrastructure of individual endosomes looked very similar in thioperamide-treated cells when compared to controls".

2-Statistical analysis is still missing for the data on the total cholesterol levels in liver shown in Figure 6B.

2nd Revision - authors' response

12 April 2019

Referee #1:

This revised manuscript is much improved and the modifications address many of the concerns raised in the prior review. A few points remain:

Fig 3B: How many cells were used for quantification? Analysis of biological triplicates with assessment of statistical significance would be most appropriate. As the data are currently presented (mean values), there is no indication of variation between samples.

We are sorry if the quantification was not clear. Immuno-EM is not easy and we tried to do the best we could. The experiment was done in Switzerland, and sent to Brisbane Australia. The data were quantified in a double-blind analysis of two sets of 16 micrographs for each condition (control cells and thioperamide-treated cells). The number of gold particles per endosome was quantified and is now shown in a scatter plot for each endosome identified without any bias in each micrograph of control and thioperamide-treated samples. I hope that this clarifies the issue.

Fig EV9B, C: How many mice/group? Include error bars and statistical analysis to support conclusion in panel C that thioperamide plus miglustat is more beneficial than either treatment alone. Include thioperamide alone group.

We are sorry about these omissions. In these experiments we used 6 mice per condition. The Legends have been corrected as well as the error bars. We have also included the thioperamide alone group.

Legend to EV9: Delete mention of rearing data, which is not included in figure.
Sorry about the confusion! The data was included in the new version of the figure.

Legend to Fig 1E: add scale bar size
The Legend has been corrected.

Page 9: LBPA quantification in NPC1 mice is shown in Fig 6B rather than 5G.
This has been corrected.

Referee #2:

The authors have addressed my queries and made an effort to further characterise thioperamide mode of action and the effects of the treatment in the mouse brain. Although, unfortunately, the effects in the brain and life span are not significant the underlying reasons for that are discussed. Overall, the results encourage further research on LBPA function and open perspectives on its use as a target for NPC treatment.

There are only two minor points that should be addressed but do not need re-review from my part.

1-Fix the apparent inconsistency on the following: While the subtitle of Figure 3 claims that Thioperamide affects endosome morphology, the text in page 5 states "analysis by electron microscopy showed that the ultrastructure of individual endosomes looked very similar in thioperamide-treated cells when compared to controls".

Oups! Again, sorry for this mistake. The title of the Legend was corrected and changed to: "Thioperamide does not affect endosome morphology or distribution".

2-Statistical analysis is still missing for the data on the total cholesterol levels in liver shown in Figure 6B.

As mentioned above (reviewer one), we are sorry about these omissions. In these experiments we used 6 mice per condition. The Legends have been corrected as well as the error bars. We have also included the thioperamide alone group

Corresponding Author Name: Jean Gruenberg

Manuscript Number: EMBOR-2018-47055